# Investigating energy production and wake losses of multi-gigawatt offshore wind farms with atmospheric large-eddy simulation

Peter Baas[1], Remco Verzijlbergh[1,2], Pim van Dorp[1], and Harm Jonker[1,3]

[1]Whiffle, Molengraaffsingel 8, 2629 JD Delft, The Netherlands
[2]Delft University of Technology, Department of Engineering Systems & Services, Jaffalaan 5, 2628 BX Delft, the Netherlands
[3]Delft University of Technology, Department of Geosciences & Remote Sensing, Stevinweg 1, 2628 CN Delft, the Netherlands

**Correspondence:** Peter Baas (peter.baas@whiffle.nl)

**Abstract.** As a consequence of the rapid growth of the globally installed offshore wind energy capacity, the size of individual wind farms is increasing. This poses a challenge to models that predict energy production. For instance, the current generation of wake models has mostly been calibrated on existing wind farms of much smaller size. This work analyses annual energy production and wake losses for future multi-gigawatt wind farms with atmospheric large-eddy simulation. To that end, one year of actual weather has been simulated for a suite of hypothetical four-gigawatt offshore wind farm scenarios. The scenarios differ in terms of applied turbine type, installed capacity density, and layout. The results suggest that production numbers increase significantly when the rated power of the individual turbines is larger, while keeping the total installed capacity the same. Even for turbine types with similar rated power, but slightly different power curves, significant differences in production were found. Although wind speed was identified as the most dominant factor determining the aerodynamic losses, a clear impact of atmospheric stability and boundary layer height has been identified. By analyzing losses of the first-row turbines, the yearly average global-blockage effect is estimated between 2 to 3 %, but it can reach levels over 10 % for stably stratified conditions and wind speeds around $8 \, \mathrm{m \, s^{-1}}$. Using a high-fidelity modeling technique, the present work provides insights in the performance of future, multi-gigawatt wind farms for a full year of realistic weather conditions.

## 1 Introduction

As part of the transition to renewable energy sources, the European offshore wind energy capacity is expanding rapidly. For example, the offshore wind energy capacity in Dutch, Belgian, Danish and German parts of the North Sea is anticipated to reach the 65 GW mark in the year 2030 and 150 GW in the year 2050 (The Esbjerg Declaration, 2022) whereas the European-wide target for offshore wind in 2050 is 300 GW (European Commission, 2020).

Ten years ago, the largest offshore wind farms had a capacity of around 500 MW. Nowadays this number has increased to 1500 MW and before the year 2030, wind farms of 4000 MW will be no exception. In fact, already today clusters of wind farms with a joint capacity of several gigawatts exist. In parallel, the wind turbines themselves increase in size. The current generation of offshore wind turbines have a nominal power of 10 to 12 MW, but this could increase to as much as 20 MW for

the year 2030. Offshore wind energy is thus entering a new phase on three levels: the total installed capacity, the size of the wind farms and the size of the individual wind turbines.

Veers et al. (2019), among others, are pointing out the need for a better understanding of atmospheric flows through wind farms. In particular the growth of wind farm size poses a challenge for models that predict energy production. The current generation of wake models has been extensively validated on wind farms in the 100 MW to 500 MW range. Using these model to make predictions for the future generation of multi-gigawatt wind farms forces them to operate well outside their validation range. This would at least add significant uncertainty to their predictions. It could therefore be argued that more

physics-based models have higher fidelity in this 'terra incognita'.

     One of such modeling techniques is Large-Eddy Simulation (LES). By numericaly integrating the filtered conservation equations of mass, momentum, temperature, and moisture, LES is able to capture the essential aspects of wind farm flow dynamics in a physically sound way. The 'global-blockage' phenomenon is a fitting example: the presence of a wind farm induces spatial gradients in the modeled pressure field, leading to forces upwind of the wind farm, thus 'informing' the flow

about the 'obstacle' ahead and causing the flow to deflect (around and/or over the wind farm).

     LES has been at the forefront of wind farm flow physics research for some time, see for example the reviews in Mehta et al. (2014), Stevens and Meneveau (2017) and Porté-Agel et al. (2020). Owing to the increase in wind turbine and wind farm scale, a number of recent studies have explored atmospheric flows through large wind farms. Maas and Raasch (2022) have studied the wake effects of a cluster of offshore windfarms in the German bight, exploring aspects like (far) wake effects,

boundary layer structure, turbulence and entrainment of kinetic energy for a selection of cases with different atmospheric stability. Verzijlbergh (2021) discussed some aspects of modelling flows through large wind farms with illustrative LES results of a 4 GW wind farm in the North Sea.

     The present work aims to explore the energy production and internal wake effects for a suite of hypothetical 4 GW offshore wind farm scenarios. The scenarios differ in terms of applied turbine type, capacity density, and layout. Furthermore, we

study how wake losses depend on atmospheric stability and we discuss the global-blockage phenomenon. Amongst others, we address questions like: how large are wake and blockage losses in 4 GW wind farms and how do these depend on wind speed, wind direction and atmospheric stability? What is the impact of turbine size and power density? How are losses distributed over the wind farms for different layouts and geometries?

     To this end, for a total of six hypothetical wind farm scenarios we simulate one year of actual weather with the GRASP

(GPU-Resident Atmospheric Simulation Platform) LES model. This is done by driving the LES with data from ECMWF's ERA5 reanalysis dataset (Hersbach et al., 2020). In this way, we obtain representative distributions of, for example, wind speed, stability, and baroclinicity in a natural way (Schalkwijk et al., 2015b). The resulting dataset can be regarded as a consistent, three-dimensional, one-year dataset of pseudo-observations of meteorological variables (including wake effects) and power production (at turbine level). As such, the present work allows for a more statistical approach to study wind farm dynamics

compared to other LES studies that mostly considered a set of idealized case studies.

     This paper is organized as follows. In Section 2 the model is introduced. The different scenarios are described in Section 3. Section 4 presents the results. After a discussion in Section 5, the Conclusions are summarized in Section 6.

## 2 Model description and simulation strategy

The model simulations are carried out with the GPU-Resident Atmospheric Simulation Platform (GRASP). GRASP is an LES code that runs almost entirely on GPUs, see Schalkwijk et al. (2012). The origin of GRASP can be traced back to the Dutch Atmospheric Large Eddy Simulation model (DALES), which is extensively described in Heus et al. (2010).

### 2.1 Governing equations

We present the most important governing equations below. More details can be found in Heus et al. (2010), Böing (2014) and Schalkwijk et al. (2015a). We follow Einstein's summation notation, with $x_1, x_2, x_3 = x, y, z$ for the coordinates and $u_1, u_2, u_3 = u, v, w$ for the wind components. The continuity equation reads:

$$\frac{\partial \rho_b u_j}{\partial x_j} = 0 \tag{1}$$

In the anelastic approximation employed in GRASP, the density $\rho_b = \rho_b(z)$ represents a base density profile depending on height only.

$$\rho_b \frac{\partial u_i}{\partial t} = -\frac{\partial \rho_b u_i u_j}{\partial x_j} - \frac{\partial \tau_{ij}}{\partial x_j} - \frac{\partial p'}{\partial x_i} + \delta_{i3} \rho_b B + \epsilon_{ij3} f_c (u_j - u_{geo,j}) + \left( \frac{\partial \rho_b u_i}{\partial t} \right)_{\text{sources}} \tag{2}$$

In the Navier-Stokes equation above, we denote buoyancy with $B$. In the buoyancy calculation a height-dependent reference temperature is used. The large-scale pressure gradient term has been written as a geostrophic wind $u_{geo}$. Further, $f_c$ denotes the Coriolis parameter and $p'$ the pressure fluctuations. The subgrid-scale turbulent stress, $\tau_{ij}$, needs to be modeled with an appropriate turbulence closure. In this study we apply the Rozema model (Rozema et al., 2015), which is a minimum-dissipation eddy-viscosity model specifically developed for anisotropic grids. As such, $\tau_{ij}$ is modeled as

$$\tau_{ij} = -2 K_m S_{ij}, \tag{3}$$

where

$$S_{ij} = \frac{1}{2} \left( \frac{\partial u_i}{\partial x_j} + \frac{\partial u_j}{\partial x_i} \right) \tag{4}$$

is (the symetric part of) the velocity-gradient tensor. The eddy viscosity/diffusivity, $K_m$, is given by

$$K_m = (c_s \Delta)^2 f(S_{ij}) \tag{5}$$

with a term containing the grid resolution, $\Delta$, a pre-factor, $c_s$, and some function of the velocity-gradient tensor. The pre-factor $c_s$ is named after the so-called Smagorinsky-constant in the traditional Smagorinsky subgrid model.

Transport of heat is described by:

$$\rho_b \frac{\partial \vartheta_l}{\partial t} = -\frac{\partial \rho_b u_j \vartheta_l}{\partial x_j} - \frac{\partial F_j^\vartheta}{\partial x_j} + S_{\vartheta_l}, \tag{6}$$

Sources/sinks of temperature are e.g. related to diabatic processes such as radiative transfer. Radiative transfer calculations are carried out off-line based on the ERA5 input profiles of the relevant variables.

We use a temperature,

$$\vartheta_l = \frac{h_l}{c_p} \tag{7}$$

that is based on moist static energy $h_l$:

$$h_l = c_p T + gz - L_v q_l - L_i q_i \tag{8}$$

This is a conserved variable for moist adiabatic ascent. Here $c_p = 1005 \text{kJ kg}^{-1} \text{K}^{-1}$ denotes the specific heat capacity of air (at constant pressure), $L_v = 2.25 \cdot 10^6 \text{J kg}^{-1}$ the latent heat of vaporization of water, and $L_i = 2.84 \cdot 10^6 \text{J kg}^{-1} \text{K}^{-1}$ the latent heat of sublimation of ice.

Transport of moisture is described by:

$$\rho_b \frac{\partial q_t}{\partial t} = -\frac{\partial \rho_b u_j q_t}{\partial x_j} - \frac{\partial F_j^q}{\partial x_j} + S_{q_t}, \tag{9}$$

where $q_t = q_v + q_l + q_i$ denotes the conserved variable total specific humidity, being the sum of vapor, liquid and ice water. Sub-grid fluxes of humidity are denoted $F_j^q$. Local sources/sinks of humidity, denoted by $S_{q_t}$, are related to microphysics.

An 'all-or-nothing' cloud adjustment scheme is used that assumes that no cloud water/ice is present in unsaturated grid boxes, while all moisture exceeding the local saturated vapor pressure is considered liquid water or ice. In addition, the Grabowski (1998) ice microphysics scheme is used. A single precipitating prognostic variable, $q_r$, is used. The partitioning towards water, snow and graupel is diagnosed with a temperature criterion. Autoconversion, the initial stage of rain drop formation, is modeled according the Kessler-Lin formulation (Khairoutdinov and Randall, 2003).

## 2.2 Boundary conditions

### 2.2.1 Large-scale NWP

In this study, the LES is coupled to ECMWFs ERA5 reanalysis dataset. As we apply periodic lateral boundary conditions, no large-scale gradients can be resolved by the LES (a model version with open boundary conditions is currently being developed). Initial conditions and large-scale ($LS$) tendencies are extracted from ERA5 by means of spatial and temporal interpolation and prescribed to GRASP as a function of height only (i.e. homogeneous over the domain). To account for the large-scale tendencies, several model terms are adjusted and/or added:

$$\rho_b \frac{\partial u_i}{\partial t} = \cdots + \epsilon_{ij3} f(u_j - u_{geo,j}^{LS}) - \rho_b u_i^{LS} \frac{\partial u_j^{LS}}{\partial x_j} - w^{LS} \frac{\partial u_i}{\partial z} + \frac{1}{\tau} \left( u_i^{LS} - \overline{u_i} \right) \tag{10}$$

And for any scalar $\phi_i$,

$$\rho_b \frac{\partial \phi_i}{\partial t} = \cdots - \rho_b u_i^{LS} \frac{\partial \phi_j^{LS}}{\partial x_j} - w^{LS} \frac{\partial \phi_i}{\partial z} + \frac{1}{\tau} \left( \phi_i^{LS} - \overline{\phi_i} \right) \tag{11}$$

The final terms of Eqs. 10 and 11 represent nudging to the large-scale model: the slab-averaged model fields $(\overline{u_i}, \overline{\phi_i})$ are nudged to ERA5 with a nudging time scale, $\tau$, of 6 h. This time scale is short enough to give the LES physics enough freedom to establish its own unique state, but short enough to make the simulation follow slow large-scale disturbances such as weather

fronts (Neggers et al., 2012). In the upper quarter of the domain the nudging time scale to ERA5 is gradually decreased (i.e. stronger nudging) towards a value of of 60 s at the domain top.

### 2.2.2 Lower boundary conditions

Over water surfaces (as in the present study), GRASP uses a prescribed surface temperature $T_s$. At the surface, saturation is assumed:

$$q_{ts} = q_{sat}\left(T_s, p_s\right) \tag{12}$$

The surface roughness lengths for momentum and heat, $z_{0m,h}$ are parameterized following the ECMWF IFS documentation ECMWF (2017):

$$z_{0m} = \alpha_m \frac{\nu}{u_*} + \alpha \frac{u_*^2}{g} \tag{13}$$

$$z_{0h} = \alpha_h \frac{\nu}{u_*} \tag{14}$$

where $\alpha$ is the Charnock parameter, taken as 0.0185. Furthermore, $g = 9.81\,\mathrm{m\,s}^2$ is the gravitational constant; $\nu = 1.5 \cdot 10^{-5}\mathrm{m}^2\,\mathrm{s}^{-1}$ is the kinematic viscosity of air, $\alpha_m = 0.11$ and $\alpha_h = 0.4$. For momentum, this parameterization follows Charnock (1955) added with viscous effects for light wind conditions.

### 2.2.3 Upper boundary conditions

At the top of the domain, we take:

$$\frac{\partial u}{\partial z} = \frac{\partial v}{\partial z} = 0; \qquad w = 0; \qquad \frac{\partial \phi_i}{\partial z} = \text{constant in time} \tag{15}$$

Fluctuations of velocity and scalars are damped out in the upper part of the domain by a sponge layer through additional forcing/source terms added to the right-hand side of the governing equations:

$$\rho_b \frac{\partial u_i}{\partial t} = \cdots - \alpha^{sp} \rho_b \left(u_i - \overline{u_i}\right) \tag{16}$$

$$\rho_b \frac{\partial \phi_i}{\partial t} = \cdots - \alpha^{sp} \rho_b \left(\phi_i - \overline{\phi_i}\right) \tag{17}$$

with $\alpha^{sp}$ a height-dependent relaxation rate (units $\mathrm{s}^{-1}$) that varies from $2.75 \cdot 10^{-3}\,\mathrm{s}^{-1}$ at the top of the domain to 0 at the height where the sponge layer starts, which is at 75 % of the domain height (i.e. the sponge layer comprises the upper quarter of the domain).

#### 2.2.4 Lateral boundary conditions

In the present setup we apply periodic boundary conditions. To prevent the recirculation of wind farm wakes, we make use of a concurrent-precursor simulation Stevens et al. (2014). This is a simulation without wind turbines that runs in parallel with the 'actual' simulation. Over a boundary region the values of the 'actual' simulation are strongly nudged towards the precursor simulation (with an adaptive nudging time scale in the order of the model time step). A schematic overview of this setup is shown in Fig. 1.

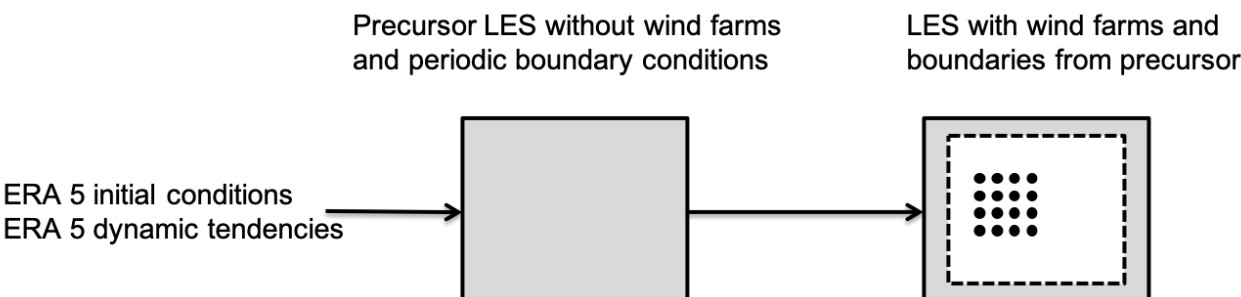

**Figure 1.** Schematic view of ERA5 boundary conditions, a precursor simulation, and a nested domain with turbines.

### 2.3 Wind turbine parameterization

Wind turbines are modeled by a so-called actuator disk model. This models each turbine as a semi-permeable disk that exerts forces on the flow that are consistent with the thrust curve of the wind turbine. In this way, wind farm wake effects are taken into account. In addition, using the turbine power curve, the turbine parameterization allows to directly model power output per turbine on a high temporal resolution. The actuator-disk model is implemented following Meyers and Meneveau (2010) 150 and Calaf et al. (2010). Within this parameterization, the total drag force exerted on the flow by a wind turbine is modelled as

$$F_t = -\frac{1}{2}\rho A C_t' \overline{M_D}^2 \tag{18}$$

where $\rho$ is the disk-averaged air density, $A = \pi R^2$ is the frontal area of the rotor and $C_t'$ the thrust coefficient based on the disk-averaged wind speed $\overline{M_D}$. Wind turbine power is given by

$$P_t = -\frac{1}{2}\rho A C_p' \overline{M_D}^3 \tag{19}$$

with $C_p'$ the disk-based power coefficient. The disk-based power and thrust coeffients are determined from the manufacturers power and thrust curves by means of an offline simulation. This additional step is required since the manufacturer curves are based on a free-stream wind speed, $\overline{M_\infty}$, a reference density, $\rho_0$, and a reference turbulent intensity, $TI_{ref}$. An additional advantage of this approach is that the turbines by definition produce the correct power and thrust for the given grid configuration.

The present implementation of the actuator-disk model has been tested extensively in operational practice and shows good performance for a wide range of numerical grid settings.

In order to quantify aerodynamic losses, we compare the energy production of the wind turbines with the production of so-called thrustless turbines. These thrustless turbines are embedded in the concurrent-precursor simulation. The disk-based power coefficients for the thrustless turbines are obtained by means of a separate offline simulation with the thrust coefficients set to 0. As a result, a power production of the thrustless turbines can be determined, but they do not exert drag on the flow. Thus, each thustless turbine produces power as if it were a single isolated turbine. Furthermore, the simulations with thrustless turbines and those with the active turbines experience exactly the same turbulent wind fields at the boundaries. As such, the difference between the production of the thrustless turbines and the active turbines is a measure of the aerodynamic loss.

## 2.4 Simulation strategy

For each of the wind farm scenarios (Sect. 3.1), the year 2015 was simulated. For this year, observations from the metmast 'Meteomast IJmuiden' were available for basic validation. The year-long simulations consist of concatenated daily simulations with a spin-up of 2 hours. For each day, GRASP is initialized at 22 h (UTC) the previous day. Model output valid between 0 and 24 h (UTC) is used for the analysis.

The model domain consists of 640 x 640 x 48 grid points. The horizontal grid spacing is 120 m, the lowest grid box has a height of 30 m. The horizontal domain size extends to 76800 m. Vertical grid stretching was applied to obtain a domain height of 3000 m (i.e. a uniform growth factor of 2.845 %). Sensitivity experiments discussed in Section 5 indicate that this domain size is sufficiently large. The model domain is centered around 52.8659° N and 3.5364° E. This corresponds to a location in the North Sea, roughly 100 km from the Dutch coast witin the planned 4000 MW wind farm IJmuiden Ver.

Compared to other LES studies (cf. Wu and Porté-Agel (2017), Maas and Raasch (2022), Strickland et al. (2022)), the horizontal resolution of 120 m is relatively coarse. This choise results from a trade-off between computational cost and accuracy and has been tested extensively in an operational setting. As such, it follows from our ambition to simulate a full year of realistic weather conditions, rather than the common approach of running a suite of targeted (idealized) case studies. To provide insights into the effect of the applied resolution, the sensitivity of the results to the grid spacing are discussed in Section 5.

As a basic validation of the model's capability to represent the local wind conditions, Figure 2a compares modeled versus observed wind speed at a height of 92 m. In this case, the modeled (horizontal) wind speed is taken from a virtual metmast placed at the location of the actual metmast. The correspondence between model and tower observations are satisfactory, with error metrics within the expected range for wind resource assessments. Figure 2b shows the distribution of the modeled 92 m wind speed, with a Weibull function fitted to the data. For comparison, grey dots indicate the distribution of the observations. Figure 2c presents the (modeled) wind rose, indicating that south-westerly winds have the highest frequency of occurrence and are generally stronger than winds from other sectors.

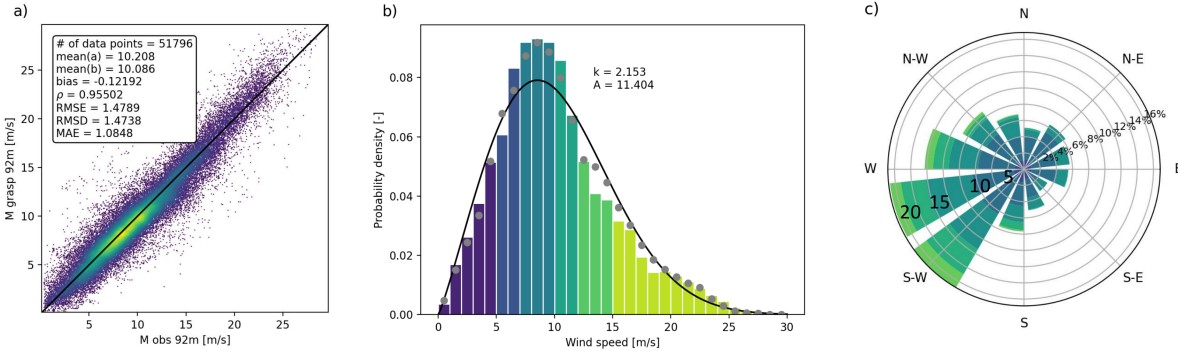

**Figure 2.** Validation results of GRASP vs. offshore tall mast IJmuiden. a) Modeled versus observed wind speed at 92 m. b) Weibull plot of GRASP 92m wind speed. Grey dots represent the observations. c) Modeled wind rose at 92 m. Colors indicate $5\,\mathrm{m\,s^{-1}}$ intervals.

## 3 Wind farms scenarios and turbine characteristics

In this Section, the six hypothetical 4000 MW wind farm scenarios and details of the applied turbine types wil be introduced.

### 3.1 Scenarios

Layouts of the six considered scenarios are given in Fig. 3. The rationale for the first five scenarios is the same: each layout consists of four sites of roughly 10 by 10 km, separated by 3 km wide corridors. Each of the four sites within each scenario has an installed capacity of approximately 1000 MW (Scenario 1 to 4). The number of turbines depends on the rated power of the applied turbine. As Scenario 5 has only half the capacity density of the other Scenarios ($5\,\mathrm{MW\,km^{-2}}$ instead of $10\,\mathrm{MW\,km^{-2}}$), each of its four sites have only half the installed capacity (i.e. 500 MW). Scenario 6 is based on the actual site boundaries of the planned IJmuiden Ver wind farm for which a tender is expected to open in 2023 RVO (2022). The installed capacity of 4000 MW corresponds to the actual plans.

### 3.2 Turbine types

To study the impact of using different turbine types while keeping the total installed power approximately the same, four different turbine types have been applied. Three reference wind turbines were used with data taken from https://nrel.github.io/turbine-models/Offshore.html: the DTU_10MW_178RWT turbine (10.6 MW, labeled as DTU10), the IEA_10MW_198RWT turbine (10.6 MW, labeled as IEA10), and the IEA_15MW_240RWT turbine (15 MW, labeled as IEA15). In addition, a 21.4 MW turbine was constructed by using the power and thrust curves from the IEA15 turbine but increasing the rotor diameter to obtain the desired rated power. Power and thrust curves for the four wind turbines are given in Fig. 4. The rated wind speed of the IEA10 is lower then the DTU10. Instead, the latter produces lower thrust. Differences between the $c_p$ and $c_t$ curves of the IEA10 and IEA15 turbines are small.

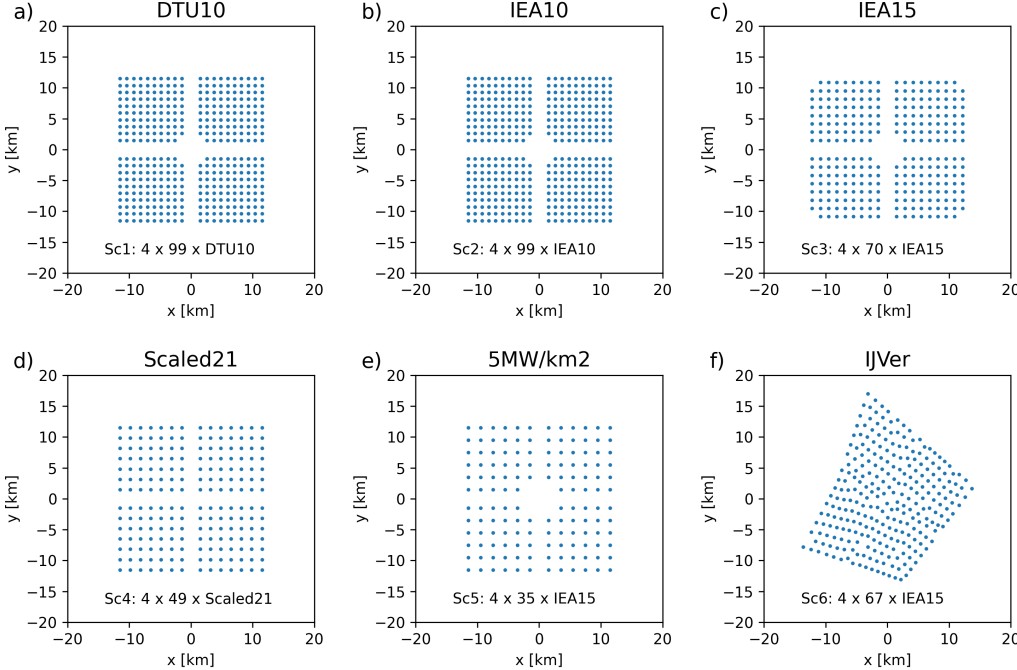

**Figure 3.** Layouts of the six wind farm scenarios. Panel titles refer to the scenario labels in Table 1. For each scenario the number and type of the applied turbine is indicated.

An overview of the scenarios and turbine characteristics is given in Table 1. The installed capacity of the first four scenarios is close to 4200 MW. For Scenario 5, with half the capacity density, this is 2100 MW. The installed capacity for the IJmuiden Ver scenario (Scenario 6) is a little lower than the other scenario's. Turbine spacing is between 5.6D and 6.2D for the 10 MW km$^{-2}$ scenarios and 8.3D for the 5 MW km$^{-2}$ scenario. These values are in the range of values that occur in existing offshore wind farms. The baseline capacity density of 10 MW km$^{-2}$ corresponds to the target set for future wind farms in the Dutch part of the North Sea. In the following, we consider Scenario 3 as a reference, for which more detailed analyses will be presented.

## 4 Results

In this Section we discuss the differences in energy production between the six scenarios. We distinguish between production of the thrustless turbines (also called 'free-stream production' or 'gross power') and the actual production ('net power'). We designate the difference between the two as the 'aerodynamic losses'. Depending on the application, we either present absolute aerodynamic losses (in MW or MWh) or relative aerodynamic losses (dimensionless) where the absolute losses are normalized with the free-stream production.

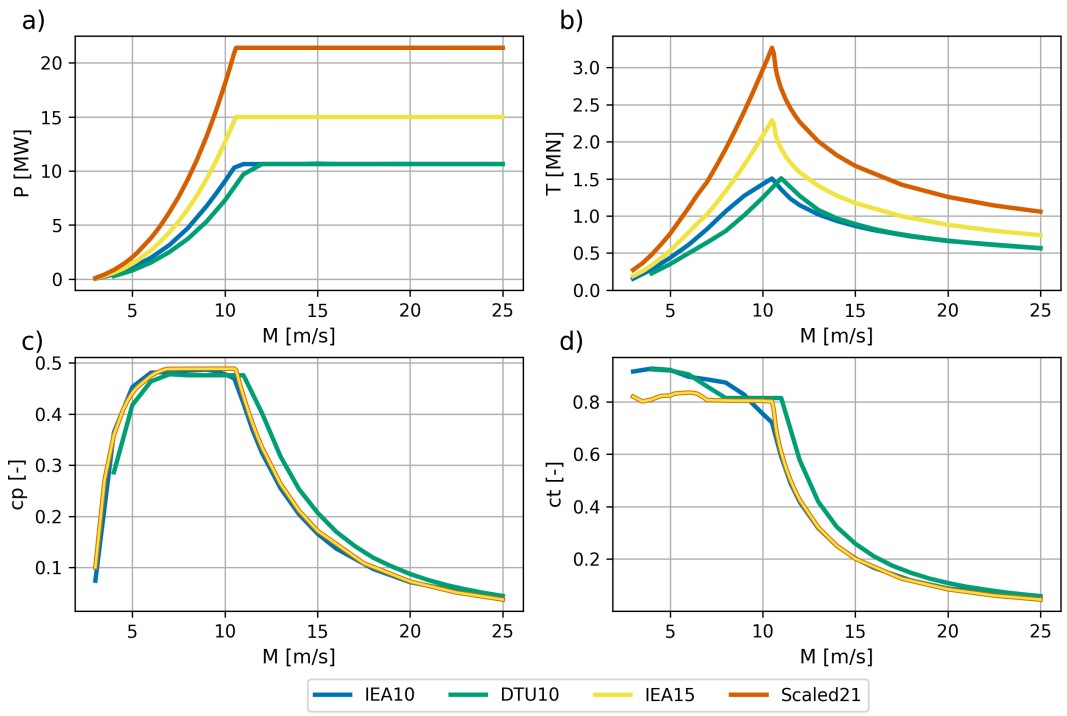

**Figure 4.** Power and thrust curves for the applied turbine types.

| Scenario | Label | Turb. type | Hub height | r | $P_{rated}$ | N | $P_{installed}$ | Spacing | Density |
|---|---|---|---|---|---|---|---|---|---|
| | | | [m] | [m] | [WM] | [−] | [MW] | | [MW km$^{-2}$] |
| 1 | DTU10 | DTU_10MW_178RWT | 119 | 89 | 10.6 | 396 | 4198 | 6.2D | 10.5 |
| 2 | IEA10 | IEA_10MW_198RWT | 119 | 98 | 10.6 | 396 | 4198 | 5.6D | 10.5 |
| 3 | IEA15 | IEA_15MW_240RWT | 150 | 120 | 15.0 | 280 | 4200 | 5.6D | 10.5 |
| 4 | Scaled21 | Scaled_21.4MW_WT | 173 | 143 | 21.4 | 196 | 4194 | 5.8D | 10.5 |
| 5 | 5MW/km2 | IEA_15MW_240RWT | 150 | 120 | 15.0 | 140 | 2100 | 8.3D | 5.4 |
| 6 | IJVer | IEA_15MW_240RWT | 150 | 120 | 15.0 | 268 | 4020 | 5.3D | 10.4 |

**Table 1.** Overview of the six scenarios, including turbines characteristics. Turbine radius is denoted by r, $P_{rated}$ denotes the turbine rated power, $P_{installed}$ the wind farm installed capacity, and N the number of installed turbines. Turbine spacing is given in number of rotor diameters, D.

After analyzing the dependence of the aerodynamic losses on the wind speed, we discuss the impact of atmospheric stability and boundary layer height. Next, losses of the first-row turbines (i.e. turbines which have no other turbines upsteam) will be considered, which gives an indication of the impact of blockage effects. We will also break down our results for bins of wind
direction. Apart from showing the impact of wind farm layout, this illustrates that for understanding directional differences a

proper separation of the wind speed effect and the stability effect is crucial. Finally, we illustrate the results with a selection of composite maps showing spatial variations of wind speed and aerodynamic losses over the wind farms.

Figure 5 presents the overall energy production and the aerodynamic losses for each of the six scenarios. The aerodynamic losses vary between 12 % and 18 % for the 4 GW wind farms whereas the 2 GW variant has losses around 6 %. Several noticable differences between the scenarios become apparent.

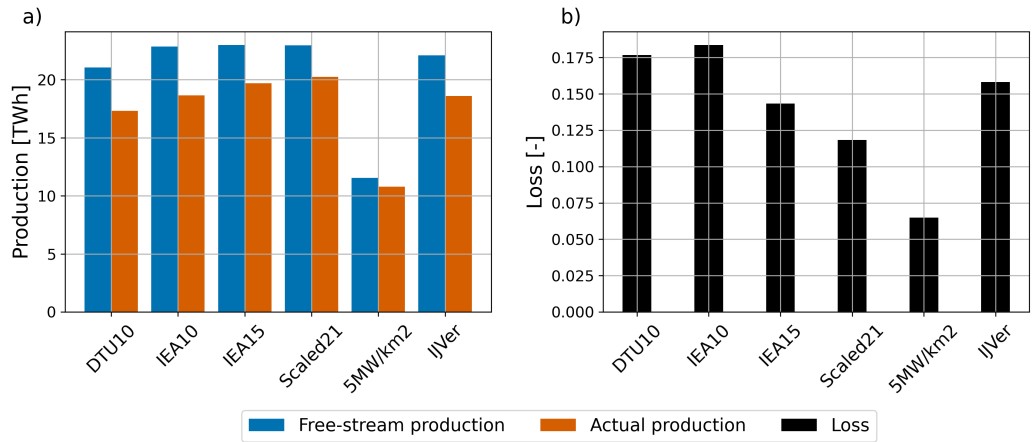

**Figure 5.** Total free stream and actual production (a) and aerodynamic losses (b) for the six scenarios.

First, although the DTU10 and IEA10 turbine have the same rated power, the actual production of the IEA10 turbine is 7.7 % larger. This significant difference is the result of higher 'free-stream' production numbers. These more than compensate for the slightly higher aerodynamic losses. Both the higher production and the higher aerodynamic losses for the IEA10 scenario can be related to a difference in the rotor diameter and a different behavior of the respective power curves (see Figure 4).

Second, while keeping the same installed power, it appear to pay-off to apply less but more powerful turbines. This is shown by comparing the IEA10, IEA15, and Scaled21 scenarios. While these three scenarios have similar 'free-stream' production, their actual production varies significantly: for IEA10, production is 5.3 % less than for IEA15, for Scaled21 the production is 2.8 % more. In terms of aerodynamic losses, this implies a reduction from 18.4 % for IEA10 to 11.8 % for Scaled21. At the same time, Table 1 indicates that the turbine spacing in terms of rotor diameters is approximately the same for these three scenarios. This suggests that the (relative) reduction of the number of turbines that is hampered by wakes of other turbines is a major factor contributing to higher production (for instance, the ratio of the number of first-row turbines over 'wake-impacted' turbines will increase (beneficial) when the total number of turbines becomes smaller).

Third, Fig. 5 illustrates the impact of varying the installed capacity per square kilometer. As expected, in the $5~\mathrm{MW\,km^{-2}}$ scenario, the free-stream production is reduced by 50 % compared to the reference IEA15 scenario. However, the actual production decreases only by 45.2 %. The aerodynamic losses decrease drastically from 14.3 % to 6.5 %.

Fourth, the results of the IJVer scenario are comparable to the IEA15 scenario. Its free-stream production is a bit less, because the installed capacity is slighty lower. Also, its aerodynamic losses are slightly higher, which is mainly related to the absence of the 3 km wide corridors (see Fig. 3).

In summary, the present results indicate that expected aerodynamic losses for a 4 GW offshore wind farm are in the range of 12 to 18 %, where the exact value is determined by the rated power of the applied turbines (or: the number of installed turbines). Moreover, turbines of the same rated capacity but different power curves may give significantly different production numbers. We emphasize that absolute numbers are related to the prevalent wind conditions in the simulated year 2015. To obtain AEP estimates that are representative for a longer period, addditional statistical postprocessing of the data are required, but this is out of the scope of the present work.

## 4.1 Wind speed-dependence of production and losses

Figure 6 considers energy production and aerodynamic loss as a function of the free-stream disk-averaged wind speed (i.e. the disk-averaged wind speed from the thrustless turbines in the concurrent precursor simulation). From left to right, the top panels represent averaged instantaneous wind farm production over the year, total energy production, and normalized cumulative production, respectively. The bottom panels show the equivalent aerodynamic losses. The results presented here are representative for the wind climate and the specific turbine design choices. A few interesting observations can be made.

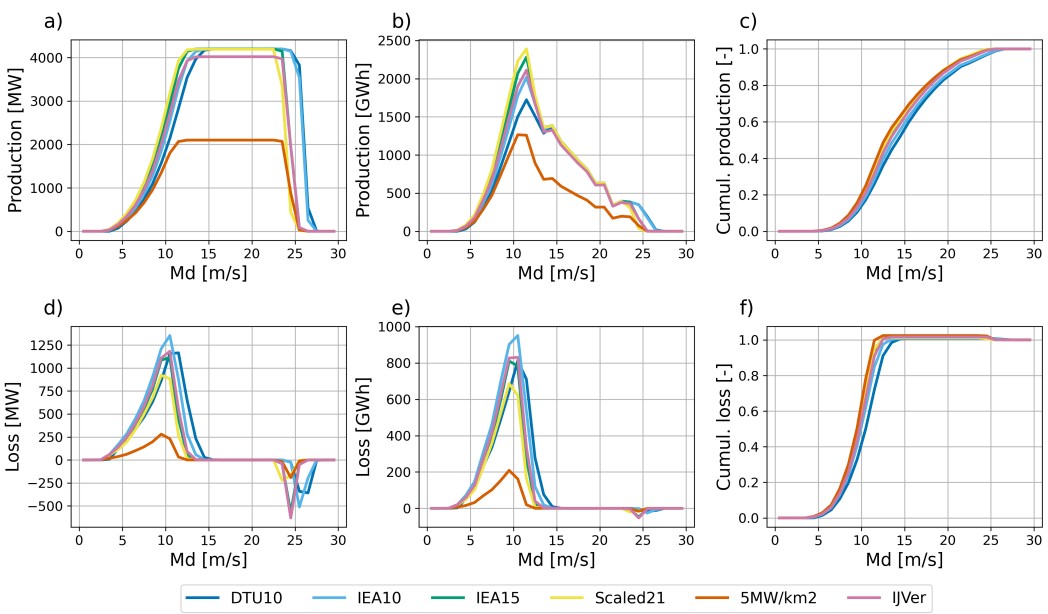

**Figure 6.** Top: year-averaged wind farm power production (a), total energy production for $1~\mathrm{m\,s^{-1}}$ bins (b), and normalized cumulative production (c) as function of the free-stream disk-averaged wind speed. Bottom: year-averaged wind farm power losses (d), total aerodynamic losses for $1~\mathrm{m\,s^{-1}}$ bins (e), and normalized cumulative losses (f) as function of the free-stream disk-averaged wind speed.

First, Fig. 6a indicates that for wind speeds stronger than $14 \text{ m s}^{-1}$ all scenarios operate at rated power. For these strong wind conditions, which generate 50 % of the total energy production (Fig. 6c), the energy content of the flow is so large that aerodynamic losses are negigible.

Second, Fig. 6d,e,f illustrate that 80 % of all aerodynamic losses occur within a narrow wind-speed range of 8 to $12 \text{ m s}^{-1}$. For lower wind speeds production and losses are low anyway, for higher wind speeds all turbines operate at (or close to) rated power. Around cut-out winds speed substantial instantaneous negative losses occur (Fig. 6d). This remarkable feature is caused by the fact that for these wind speeds, as a result of subtle wake effects, the number of power-producing turbines in the simulations with actual (thrust-generating) turbines is larger than in the simulations with the thrustless turbines. As the frequency of occurrence of these specific wind conditions is low, the impact of this effect on the integrated losses is small (Fig. 6e).

Third, the total energy production peaks around a wind speeds of approximately $12 \text{ m s}^{-1}$. This can be understood by interpreting the total energy production as a function of wind speed as a multiplication of the wind speed probability density (Fig. 2b) and the power curves.

Differences between the six scenarios are small. They are consistent with the total production numbers of Fig. 5 and can be explained by the differences in the turbine power curves (Fig. 4).

## 4.2 Impact of stability and boundary layer height

In this sub-section, we attempt to isolate the impact of stability and boundary layer height from the impact of the wind speed itself. For clarity reasons, mainly results for the IEA15 reference scenario are presented.

The impact of atmospheric stability on wake losses of wind farms has been widely reported in the scientific literature, see e.g. Stevens and Meneveau (2017). As a stability parameter, we choose the bulk Richardson number, $R_b$, over the rotor blade of the IEA15 turbine, i.e. between heights of 270 and 30 m:

$$R_b = \frac{g}{\vartheta_l} \frac{\Delta z \Delta \vartheta_l}{(\Delta u)^2 + (\Delta v)^2}. \tag{20}$$

Values of $R_b$ are taken from the precursor simulation. As such, they represent free-stream (or undisturbed) conditions. We consider three classes of stability, separated by the 33.3[th] and 66.6[th] percentiles of the year-round distribution of $R_b$, which have values of -0.04 and 0.44, respectively. As such, the stability class with the 33.3 % of lowest $R_b$ values represents convective conditions, while the class with the 33.3 % of highest $R_b$ values represents significantly stable conditions. The class of intermediate stability contains neutral conditions, but is dominated by weakly-stratified conditions.

Figure 7 presents the aerodynamic losses as a function of the free-stream disk-averaged wind speed for the three stability classes for the IEA15 scenario. For a wide range of wind conditions, the impact of stability is small. However, just in the wind speed range where most of the actual losses occur, a clear impact of stability is observed. Here, for the most stably-stratified conditions, relative losses are roughly 10 percentage points larger than for convective conditions. For higher wind speeds, losses quickly reduce to zero, irrespective of stability. For lower wind speeds, absolute losses (and production) are small.

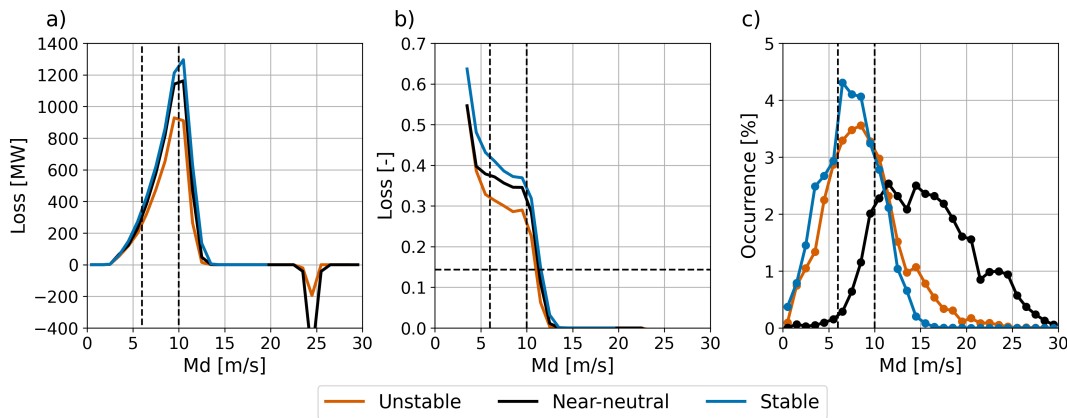

**Figure 7.** Combined effect of wind speed and stability on wind farm aerodynamic losses. a) Power losses in MW. b) Relative aerodynamic losses. c) Fequency of occurrence of the three stability classes. Dashed lines at 6 and $10 \mathrm{~m\,s^{-1}}$ indicate the wind speed interval for which the aerodynamic losses are relatively constant.

| Scenario | Unstable | Neutral | Stable |
|----------|----------|---------|--------|
| DTU10 | 0.29 | 0.37 | 0.45 |
| IEA10 | 0.34 | 0.41 | 0.48 |
| IEA15 | 0.29 | 0.35 | 0.38 |
| Scaled21 | 0.26 | 0.29 | 0.32 |
| MW/km2 | 0.14 | 0.18 | 0.20 |
| IJVer | 0.32 | 0.36 | 0.41 |

**Table 2.** Relative aerodynamic losses per scenario for free-stream disk-averaged wind speeds between 6 and $10 \mathrm{~m\,s^{-1}}$, for the three stability classes.

The strong dependency of aerodynamic losses on the wind speed may easily obscure an analysis to the impact of stability. The relevant wind speed range for considering the impact of stability seems to be between 6 and $10 \mathrm{~m\,s^{-1}}$. This narrow range of wind speeds is characterized by near-constant relative losses, which allows for a fair comparison between stability conditions. As can be seen in Fig. 4, this specific wind speed range coincides with the power and thrust curves being at their maximum. In the following, to indicate any impact of stability, we include only data for which the wind speed is between 6 and $10 \mathrm{~m\,s^{-1}}$.

Table 2 summarizes the relative aerodynamic losses for all six scenarios for disk-averaged wind speeds between 6 and 10 $\mathrm{m\,s^{-1}}$. Considerable differences between scenarios exists: the higher the overall aerodynamic losses (cf. Fig. 5), the larger the impact of stability. For example, the impact of stability is clearly smaller for the Scaled21 and $5 \mathrm{~MW\,km^{-2}}$ scenarios.

Te summarize, the impact of stability is only significant for a small range of wind speed conditions. However, it is exactly this range that is also most relevant for aerodynamic losses.

Apart from stability, other LES wind farm studies indicate that the boundary layer height, $h$, may have substantial impact on wakes and wind farm production (e.g. Maas and Raasch (2022)). Here, we examine the influence of the boundary layer height on the aerodynamic losses for the IEA15 scenario. To that end, we diagnosed the boundary layer height from model output of the precursor simulation (undisturbed conditions). We take $h$ as the height at which the magnitude of the momentum flux becomes less than 5 % of its surface value.

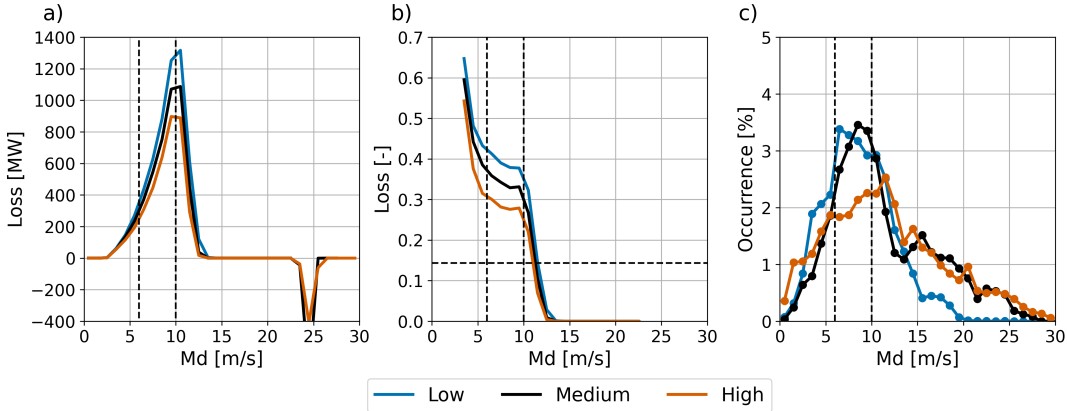

**Figure 8.** Combined effect of wind speed and boundary layer height on wind farm aerodynamic losses. a) Power losses in MW. b) Relative aerodynamic losses. c) Fequency of occurrence of the three boundary layer height classes. Dashed lines at 6 and $10\,\mathrm{m\,s^{-1}}$ indicate the wind speed interval for which the aerodynamic losses are relatively constant.

We distinguish three classes of $h$, separated by the $33.3^{\text{th}}$ and $66.6^{\text{th}}$ percentiles of the year-round distribution of $h$, which have values of 341 and 955 m, respectively. Figure 8 presents the aerodynamic losses as a function of the free-stream disk-averaged wind speed for the three classes of boundary layer height. The results show remarkable resemblance with the stability analysis (Fig. 7). Also here, the impact is mostly confined to the wind speed range between 6 and $10\,\mathrm{m\,s^{-1}}$. Within this range, aerodynamic losses for shallow boundary layers are clearly (around 10 percent points) higher than for deep boundary layers.

Obviously, stability and boundary layer height are related. This is illustrated in Table 3, which shows the simultaneous occurrence of the three classes of stability and boundary layer height. Especially the shallow boundary layers clearly coincide with stably-stratified conditions.

### 4.3 First-row losses

As with any obstacle placed in a flow, wind farms will have an impact on the flow itself. The air will tend to flow around and over the 'obstacle', and in front of the wind farm a reduction in wind speed is expected. This will lead to a reduction of power production of the turbines that are not in the wake of other turbines (i.e. located at the 'first-row'). This phenomenon is know as the global-blockage effect Bleeg et al. (2018). As the wind speed reduction will propagate to downstream ('waked') turbines, separating the blockage effect from wake effects is virtually impossible. This is especially true for observations and

|         | Low  | Medium | High | Total |
|---------|------|--------|------|-------|
| Unstable | 0.3 | 13.9 | 19.1 | 33.3 |
| Neutral | 6.1 | 15.8 | 11.4 | 33.3 |
| Stable | 24.6 | 5.2 | 3.6 | 33.3 |
| Total | 31.0 | 34.9 | 34.1 | 100.0 |

**Table 3.** Contingency table showing the simultaneous frequency of occurrence (in %) of the three classes of stability (unstable, neutral, stable) and boundary layer height (low, medium, high).

physically-based modeling studies like LES. Therefore, in this study we focus on losses of the first-row turbines, which can be interpreted as a conservative estimate for the blockage effect.

We determine the first-row losses as follows: given the wind direction, for each time step we verify if any other turbines are located within a 60 degree wide sector opposite to the flow direction. If this is not the case, a turbine is classified as a first-row turbine for that particular timestep.

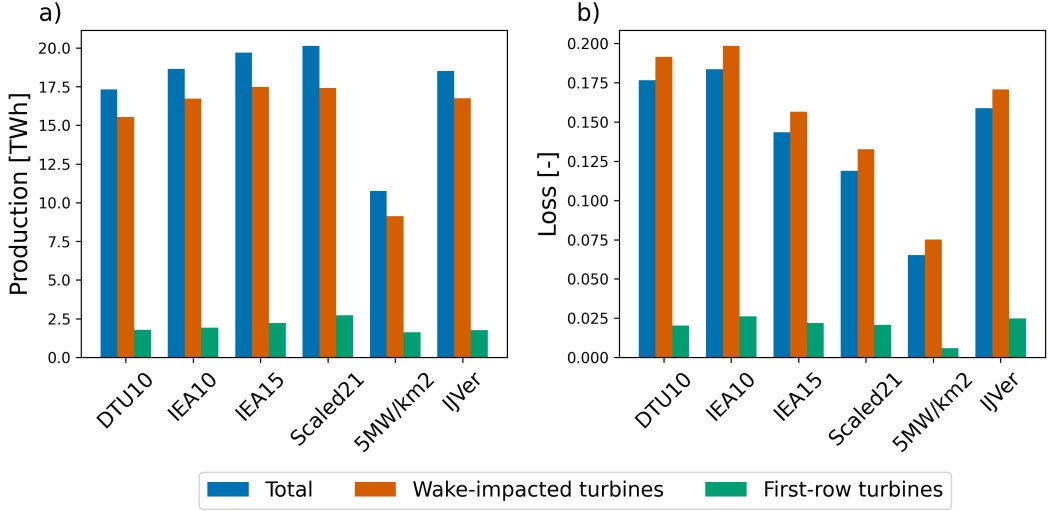

**Figure 9.** a) Total production, production of 'wake-affected' turbines, and production of first-row turbines for all six scenarios. b) Aerodynamic losses.

Figure 9 presents the year-round production numbers and relative aerodynamic losses for the first-row turbines and all other ('waked') turbines. The actual production of the first-row turbines is between 2 and 3% lower than their corresponding thrustless (or free-stream) production. Although the applied definitions and metrics can be discussed, these values are not

inconsistent with values of the blockage effect reported in literature (e.g. Wu and Porté-Agel (2017), Allaerts et al. (2018), Bleeg et al. (2018), Schneemann et al. (2021)). Consistently, the losses of the non-first-row, or other turbines, are a bit higher than the overall losses.

As with the overall aerodynamic losses above, we can also assess the impact of both wind speed and stability on the first-row losses. Figure 10 shows that, consistent with the above results (e.g. Fig. 6), also the first-row losses are negligible for wind speeds over $12\,\mathrm{m\,s^{-1}}$. Interestingly, the first-row wind speed deficit with respect to free-stream conditions continues towards much higher wind speeds. The majority of the first-row losses occur for wind speeds between 6 and $10\,\mathrm{m\,s^{-1}}$. Values range from 4 % in convective conditions to 8 % in the most stable conditions. The corresponding first-row wind speed deficits vary from approximately 0.12 to $0.30\,\mathrm{m\,s^{-1}}$. Relative first-row losses are even higher for wind speeds below $6\,\mathrm{m\,s^{-1}}$, but these are less relevant in an absolute sense (not shown).

We conclude that first-row losses are on average between 2 and 3 %. However, for the wind speed range where most of the losses occur these numbers can be more than twice as high. Also, first-row losses are significantly larger for stably-stratified condions (cf Strickland et al. (2022)).

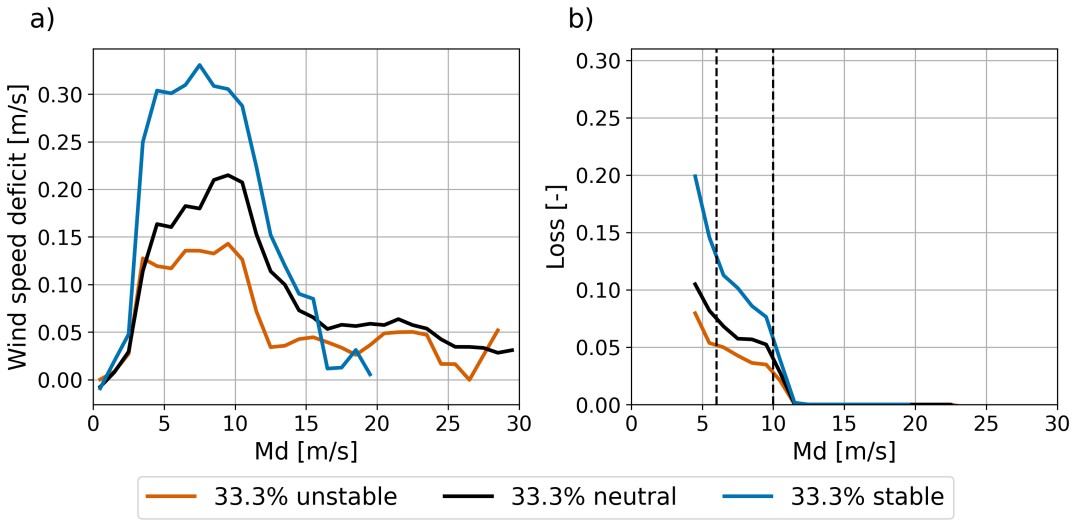

**Figure 10.** The reduction of the first-row 140 m wind speed compared to the free stream wind speed (a) and the relative aerodynamic losses of first-row turbines (b) as a function of wind speed and stability.

## 4.4 Directional effects

An analysis of aerodynamic losses per wind direction reveals how the respective impacts of wind speed and stability are entangled. Moreover, it shows the impact of difference in layout and geometry of the wind farm scenarios.

Figure 11 shows energy production and aerodynamic losses as function of the wind direction. The first element that stands out is the overwhelming dominance of the contribution of southwesterly winds to the total energy production. This is the cumulative effect of both the higher frequency of occurrence and the generally stronger wind speeds (cf. Fig. 2c), in combination with strongly non-linear character of the turbine power curves.

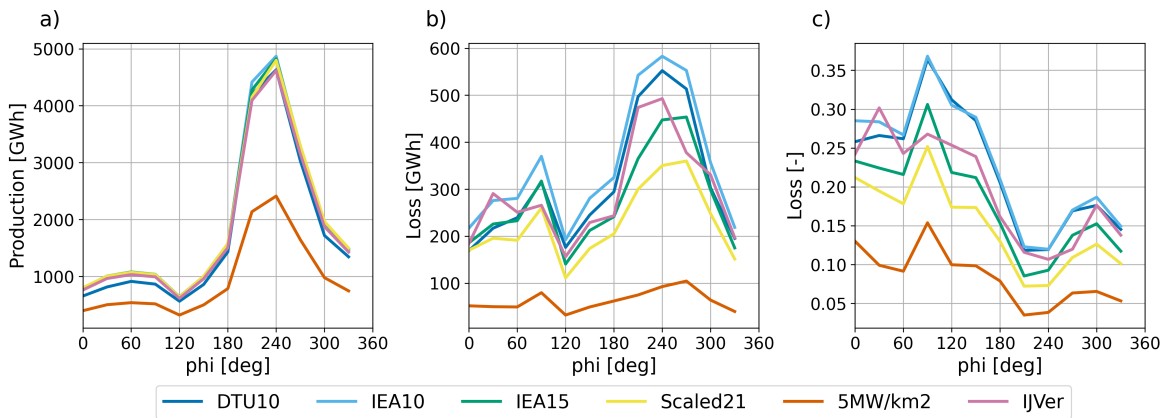

**Figure 11.** Directional dependence of total energy production (a), absolute aerodynamic losses (b) and relative losses (c) for the six scenarios.

Figure 11b,c show that while the absolute losses are largest for southwesterly direction, the relative losses are much higher for easterly directions. From this Figure, it cannot be determined if the difference in relative losses is mainly a wind speed effect or that stability is important here. Interestingly, the five hypothetical layouts closely follow the same pattern, but the IJVer scenario behaves differently. Comparison with Fig. 3 suggests that this difference is related to the different layout of the IJVer scenario: while other scenarios form north-south, west-east facing squares, the IJVer layout is significantly rotated (but still resembling a clear 'square-like' shape). Inspection of Fig. 11b,c indicates that aerodynamic losses are higher/lower when the flow is directed towards the faces/corners of the wind farm layouts.

For two of the scenarios, IEA15 and IJVer, Fig. 12 breaks down the directional losses to stability and wind speed. The top panels (a, b) present the relative aerodynamic losses for the three stability classes defined above, irrespective of the wind speed. Losses for stably-stratified conditions are largest, but also the losses for convective conditions are large. Because of generally higher wind speeds (i.e. lower thrust coefficients), the losses for the near-neutral class are much smaller, even when omni-direction numbers are considered (not shown).

As a next step, the bottom panels of Fig. 12 present stability dependant losses like before, but now only including wind speeds between 6 and 10 $\mathrm{m\,s^{-1}}$. By doing so, a clear organization of the data occurs, with the lowest losses occurring for convective conditions and the highest losses for the most stably-stratified conditions. Moreover, a clear directional pattern is revealed, in particular for the IEA15 scenario, with much higher losses when the flow is directed to the sides of the wind farm and lower losses when the flow faces the corners of the wind farm. This pattern is clearly visible for all three stability classes. For the IJVer scenario the directional pattern is more obscured.

In summary, Fig. 12 demonstrates that an assessment of the impact of stability on wind farm losses is not straightforward. It can only be isolated if the data are also conditioned over a particular, carefully selected wind speed range. This is because both the turbine thrust curves and the stability are depending on the wind speed, but in a different way. To avoid the impact of wind

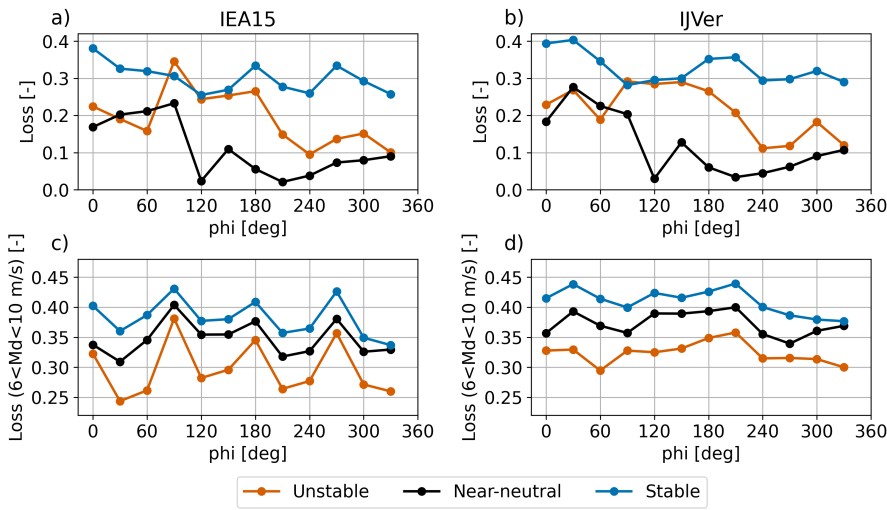

**Figure 12.** Directional dependence of total aerodynamic losses for different stability classes for the IEA15 (a,c) and IJVer (b,d). The top panels (a,b) are based on all data, the bottom panels (c, d) only include wind speeds between $6\,\mathrm{m\,s^{-1}}$ and $10\,\mathrm{m\,s^{-1}}$.

speed as much as possible, this range should not be too broad, as small differences in wind speed can have a large impact on both absolute and relative aerodynamic losses (Fig. 7).

## 4.5 Spatial patterns

So far, we only considered power production and aerodynamic losses for the wind farms as a whole. In the following section,

we consider spatial variations in wind speed, power production, and aerodynamic losses over the wind farms. By breaking down the dataset into bins of wind direction, wind speed, and stability classes, the impact of different atmospheric conditions can be examined. A selection of composite maps of aerodynamic losses, wind speed, and the ratio of actual to free stream wind speed (taken from the precursor simulation) are presented.

Figure 13 shows aerodynamic losses, mean wind speed and velocity deficit compared to the free-stream flow for the IEA15

scenario, averaged over the entire year and all wind directions. Losses vary from around 6 % for turbines located at the outside of the wind farm, to 20 % for turbines in the interior of the wind farm. The dominance of stronger southwesterly winds is reflected in lower losses in the southwestern part of the wind farm and a clear asymmetry in the composite wind fields. The impact of the wind farm on the year-round, omni-directional wind field is in the order of 20 km, after which a velocity deficit of less than 1 % is observed.

For comparison, Fig. 14 shows the results for the $5\,\mathrm{MW\,km^{-2}}$ scenario. As expected, losses are much lower than compared to IEA15, which has a capacity density of around $10\,\mathrm{MW\,km^{-2}}$. This is the combined effect of larger distance between the turbines and the fact that only half the number of turbines is involved. The impact on the mean wind field and the corresponding

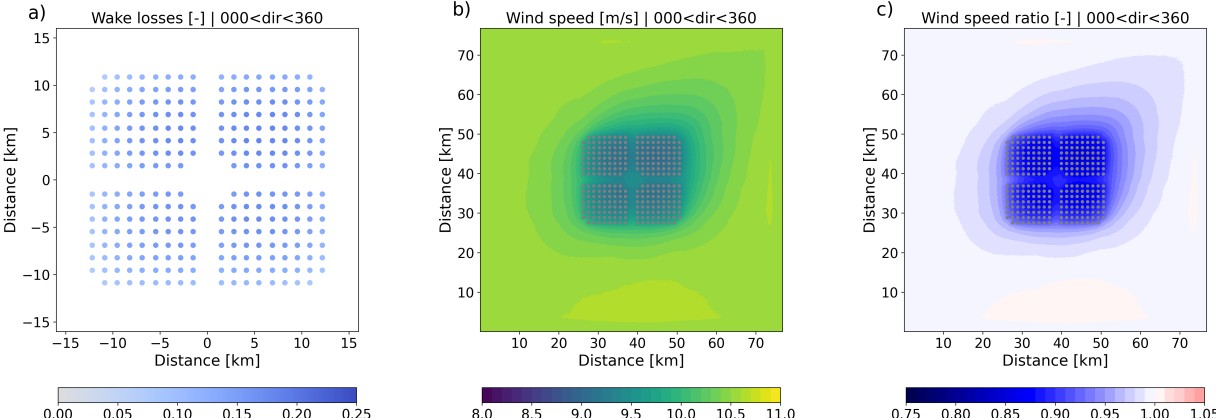

**Figure 13.** Aerodynamic losses (a), mean 140 m wind speed (b), and ratio of actual to free stream wind speed (c) for the IEA15 scenario (including all data).

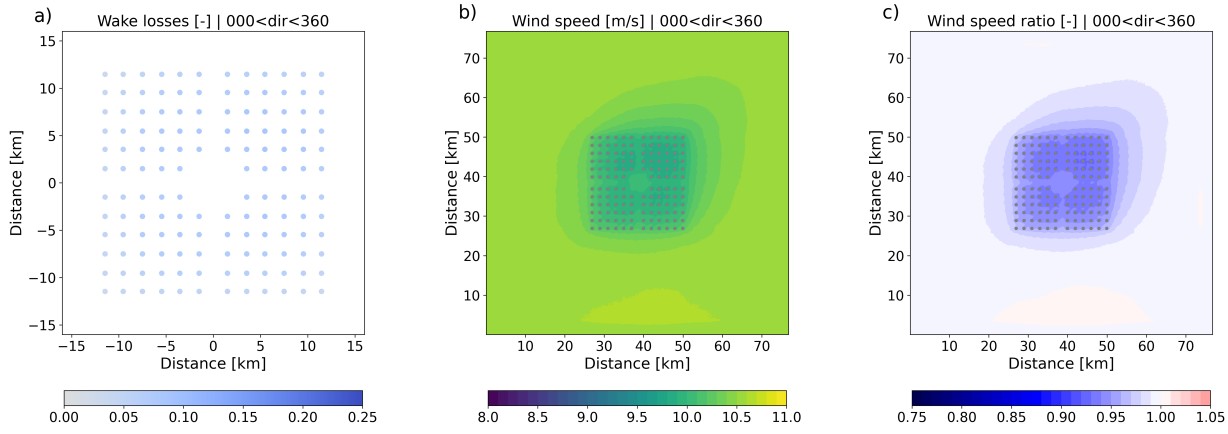

**Figure 14.** As Fig. 13, but for the 5 MW km$^{-2}$ scenario (including all data).

velocity deficit is smaller as well: in the centre of the wind farm the velocity deficit is 6 %, compared to 12 % in the 10 MW km$^{-2}$ case.

Figure 15 presents composite maps for the IEA15 scenario again, but now only including data with a wind direction between 15 and 45 degrees. In this case, a clear wake is visible, which is still present as the flow reaches the southern edge of the domain. Clearly, for studying wake lengths behind wind farms of this size, much larger domains are required than the present 80 kilometers. Upstream, the wind speed is already reduced before the flow reaches the wind farm, which signals the presence of blockage. Along the sides, a clear flow acceleration is visible. The distribution of aerodynamic losses over the wind farm

shows interesting patterns. Although not in the wake of any other turbines, the first-row turbines in the northeastern corner of the wind farm produce 10 % less power than their 'thustless' equivalents. On the other hand, the turbines in the southeast profit from the flow acceleration around the wind farm and produce up to 5 % more power than if they would have operated in isolation.

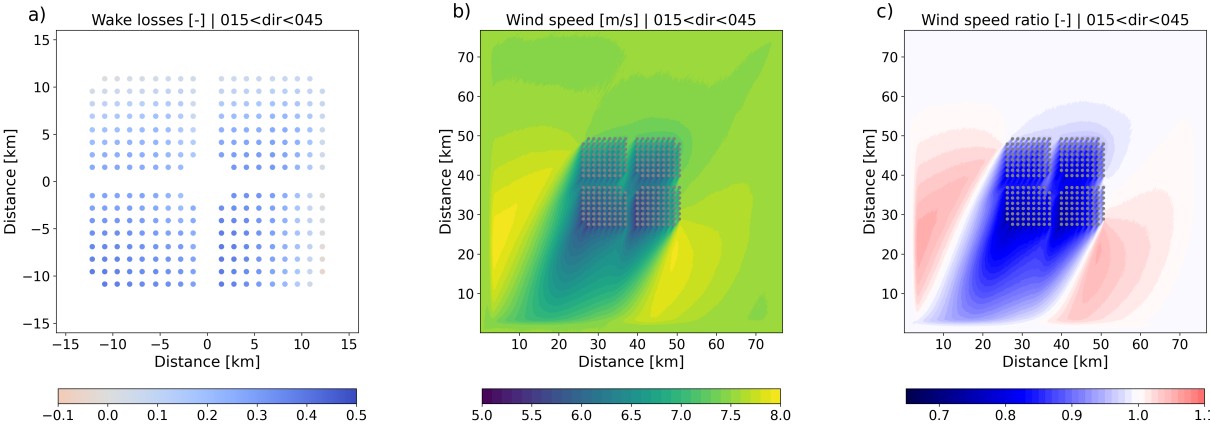

**Figure 15.** As Fig. 13, but only including wind directions between 15 and 45 degrees.

Comparison of Fig. 15 with Fig. 16 clearly illustrates the difference of the flow being oriented to the corner of the wind farm or directly towards the one of the sides. In case of the latter, the numbers of turbines that are facing undisturbed conditions (apart from blockage effects) is much less, resulting in larger aerodynamic losses (cf. Fig. 12).

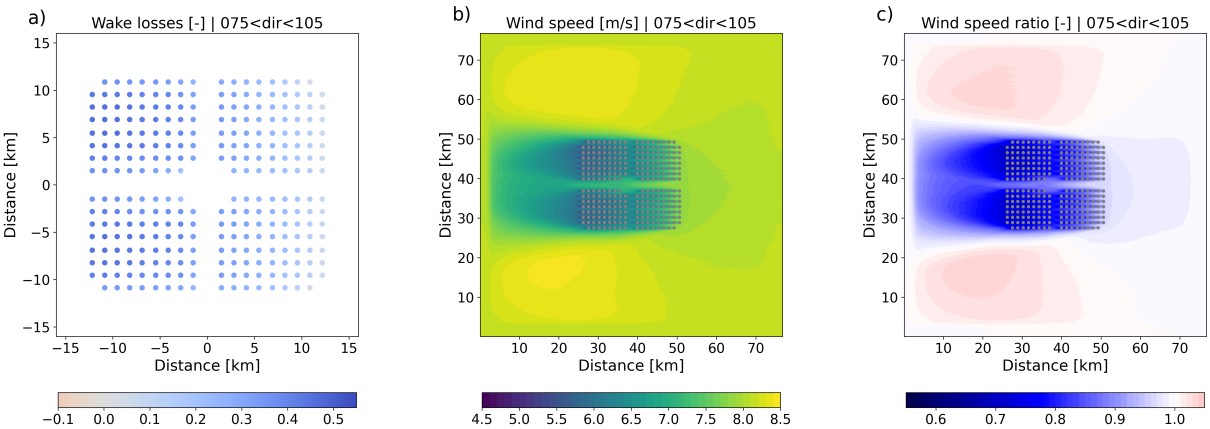

**Figure 16.** As Fig. 13, but only including wind directions between 75 and 105 degrees.

The different layout of the IJVer scenario, makes that aerodynamic losses are relatively low for easterly flow (Fig. 17). Also here, flow acceleration around the wind farm leads to increased production for, in this particular case, the northernmost turbines.

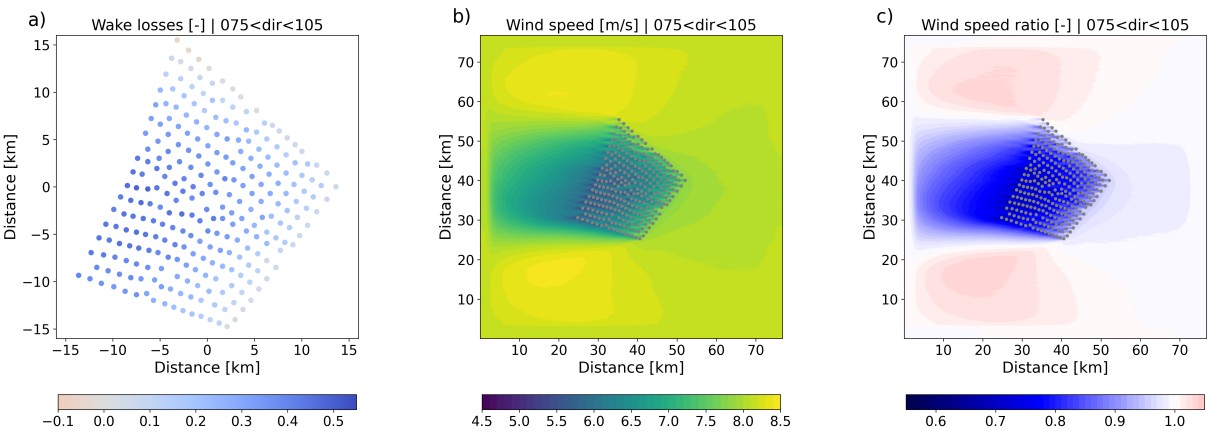

**Figure 17.** As Fig. 13, but for the IJVer scenario, only including wind directions between 75 and 105 degrees.

Finally, Fig. 18 and Fig. 19 illustrate the impact of convective and stable conditions, respectively. To enable a 'fair' comparison, only conditions with wind speeds between 6 and 10 m s$^{-1}$ are included. As shown before, in this wind speed range the aerodynamic losses are much higher than average. In stably-stratified conditions deeper wakes occur that extend further downstream. Also, the wind speed reduction upstream of the wind farm is larger in stable conditions. This is reflected in larger first-row losses compared to convective conditions. Moreover, going deeper into the wind farm, losses increase faster for stable than for convective conditions: near the southern edge of wind farm turbine losses increase to around 60 % for stable conditions, while being confined to approximately 40 % in convective conditions.

## 5   Discussion and sensitivity study

To assess production numbers and aerodynamic losses for a suite of hypothetical 4 GW offshore wind farms, a full year of simulations with the LES model GRASP have been performed. Even while GRASP has a relatively high computational performance due to its implementation on GPU's, the computational costs of the simulations are significant. That is to say, in order to enable the atmospheric simulations of large wind farms covering an entire year, the configuration of both the model grid and the domain need to be carefully selected to limit computational cost while maintaining physically sound results.

Because the applied horizontal grid spacing of 120 m might be considered coarse for an atmospheric LES model and/or for the actuator disk model that is used, we consider an asessment of the sensitivity of the modeling results appropriate. Therefore, additional simulations have been performed in which we varied the resolution, the prefactor of the subgrid model (governing

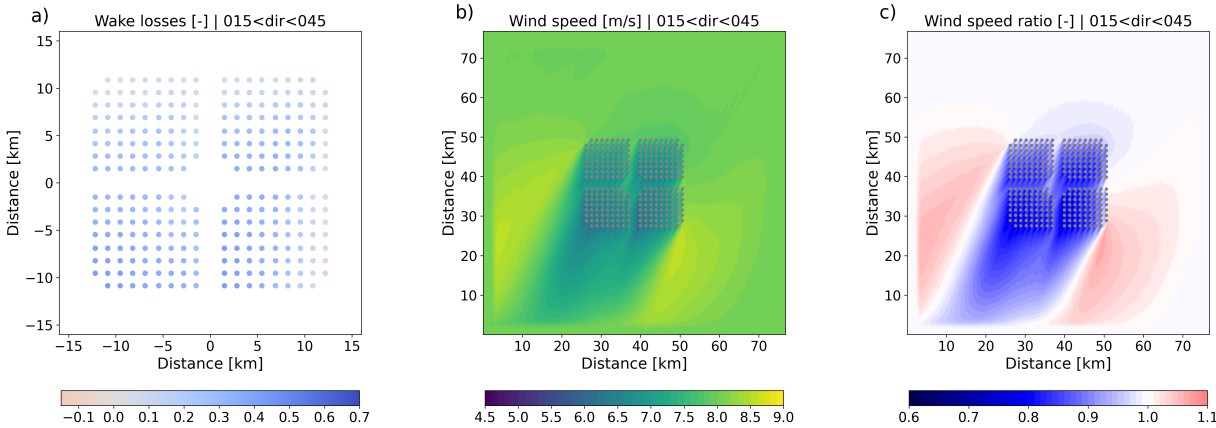

**Figure 18.** As Fig. 13, but only including convective conditions, wind speeds between 6 and $10 \, \mathrm{m \, s^{-1}}$, and wind directions between 15 and 45 degrees.

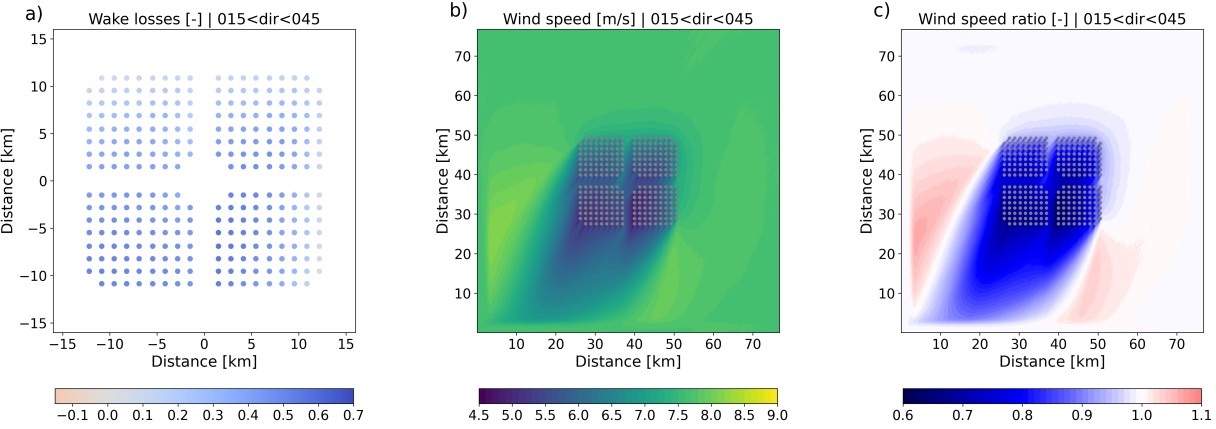

**Figure 19.** As Fig. 13, but only including stable conditions, wind speeds between 6 and $10 \, \mathrm{m \, s^{-1}}$, and wind directions between 15 and 45 degrees.

the magnitude of the subgrid-scale diffusion), and the domain size (both height and horizontal extent). The sensitivity experiments were performed on a smaller domain of $30720 \, \mathrm{m}$. A wind farm of around $770 \, \mathrm{MW}$ was included. To assess if relative differences between scenarios remained the same, each sensitivity experiment was carried out twice: one time with 72 of the IEA10 turbines (regular 9 by 8 array, spacing of 5.6D), and one time with 36 of the Scaled21 turbines (regular 6 by 6 array, spacing of 5.8D). The sensitivity experiments were not run for the entire year, but for a representative subset of 100 days. The

100 days were selected by a k-means clustering method based on the daily mean of the longitudinal and latitudinal components of the ERA5 100 m wind.

Specifically, the following sensitivity experiments have been performed:

- REF: reference simulation on a 30720 m domain of 3000 m height. The horizontal grid-spacing was 120 m, the height of lowest grid box 30 m (as in the main simulations). Number of grid points: 256 in the horizontal, 48 in the vertical.

- HR: as REF but with the horizontal grid-spacing set to to 60 m. To keep the domain size the same, the number of grid points in the horizontal was increased to 512.

- $C_s$: as REF but with the $c_s$ prefactor of the subgrid-scale eddy-diffusivity increased by 50 % (see equation 5).

- $2L_x$: as REF but with a twice as large horizontal domain of 61440 m using 512 grid points in both horizontal directions.

- $2L_z$: as REF but with the domain height increased to 6000 m using 68 vertical levels.

- $5L_z$: as REF but with the domain height increased to 14500 m using 96 vertical levels.

Modifying the modeling setup may impact both the ambient conditions (which will change the thrustless production numbers) and the interaction between the turbines of the wind farm (changing the aerodynamic losses). Fig. 20 presents the relative differences between each sensitivity experiment and the REF experiment. Differences in free stream (thrustless) production are mostly less than 1 %. The same is true for the actual production numbers. Naturally, the aerodynamic losses of the sensitivity experiment are smaller than in the main simulations as the installed capacity is smaller.

Increasing the resolution from 120 to 60 m leads to slighty lower aerodynamic losses. This is expected as at finer resolution turbine wakes are more accurately resolved and less smeared out over the grid. Still, the impact is relatively small, especially given the factor of eight difference in computational cost (number of points in the domain and a 50 % reduction in the model time step). Increasing the prefactor of the subgrid-scale eddy-diffusivity $c_s$ by 50 % increases the subgrid-scale diffusion, logically leading to a decrease in resolved fluctuations. As shown by the $c_s$ experiment, the impact on the aerodynamic losses is small. A common way to assess the validity of a large-eddy simulation is to consider the fraction of resolved turbulence. In our main simulation, the resolved fraction of the momentum flux is larger than 80 % for 70 % of the time (at a height of 150 m, which is the hub-height of the IEA15 turbine). For stably-stratified conditions the contribution of the subgrid-scale fluxes is larger, but situations where all turbulent fluctuations dissappear are rare. In practice, a relatively large (fractional) subgrid-scale contribution may have limited effect, as the absolute values of the turbulent fluxes are small.

The sensitivity experments were performed for two contrasting wind farm scenarios in order to verify the robustness of the relative differences between the scenarios. Figure 20b indicates that while the aerodynamic losses may change a bit between the sensitivity experiments, the two scenarios show similar patterns. This gives confidence in the comparison between different scenarios in Section 4.

It can be argued that the impact of the sensitivity experiments as discussed above is masked by the fact that for wind speed above 14 m s$^{-1}$ (related to 50 % of the production) losses are negligble anyway (cf. Fig. 6). Therefore, Fig. 21 presents relative

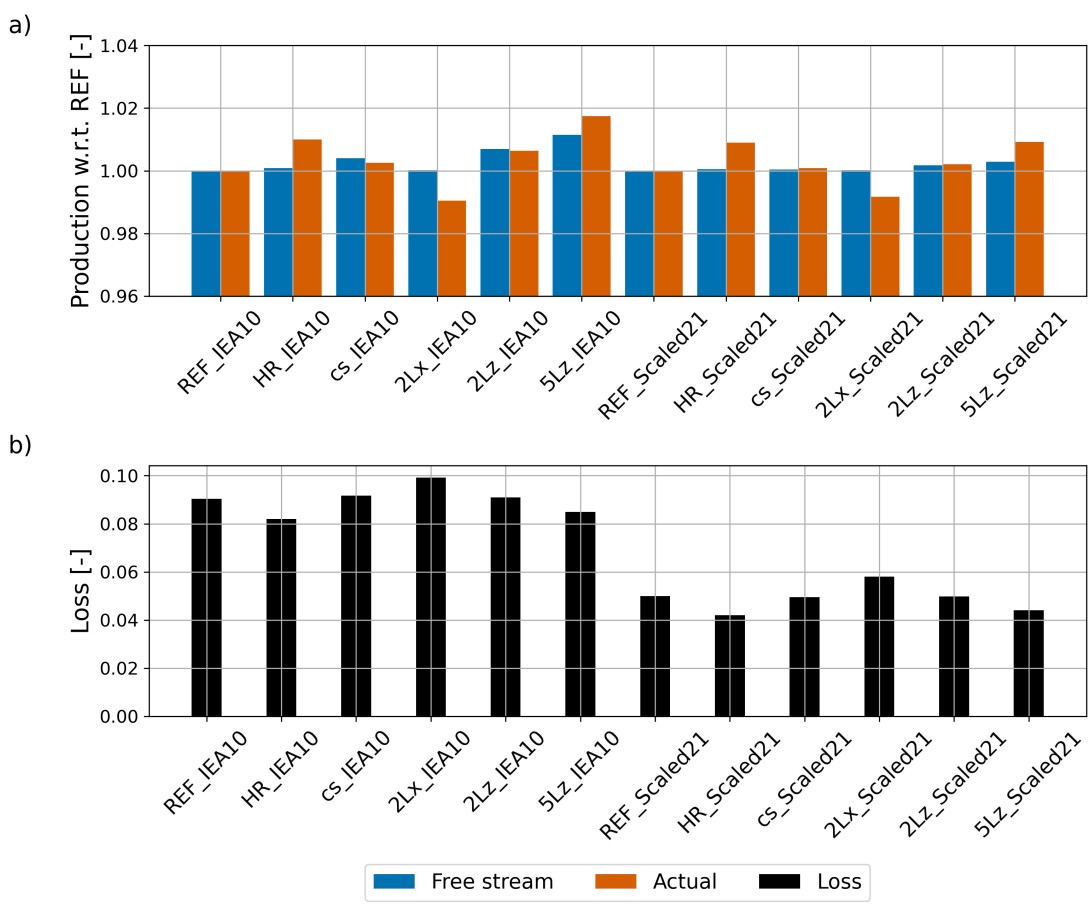

**Figure 20.** Free stream and actual production of the sensitivity experiments with respect to REF (a) and the corresponding aerodynamic losses (b).

aerodynamic losses for disk-averaged wind speeds between 6 and 10 $\mathrm{m\,s}^{-1}$. As expected, aerodynamic losses for this specific wind speed range are higher than the overall losses, as are the differences between the scenarios. Still differences with the REF simulations remain within reasonable limits. Presented numbers are for the three stability classes defined above. The differences between the stability classes are similar for the different sensitivity experiments. This gives confidence in the analysis on the impact of stability in the main results section.

Increasing the horizontal and vertical extent of the domain both have a modest impact on the production numbers and aerodynamic losses. With a twice as large horizontal domain the aerodynamic losses become slightly higher. This may be related to the additional space around the wind farms, reducing the tendency of the flow to accelerate along the wind farm's edges.

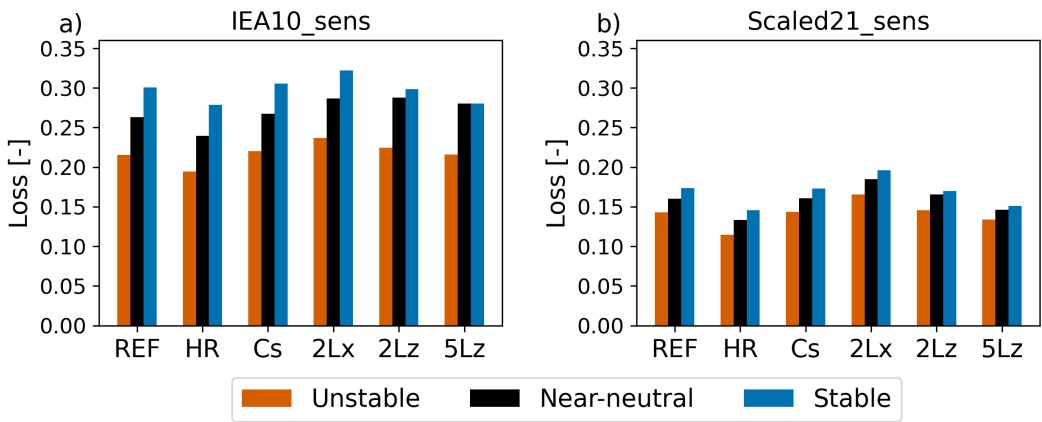

**Figure 21.** Stability-dependent aerodynamic losses for disk-averaged wind speeds between 6 and 10 $\mathrm{m\,s^{-1}}$ for the IEA10 (a) and Scaled21 (b) sensitivity experiments.

Recently, several LES wind farm studies have argued that for a proper modeling of flow through large wind farms large domain heights (usually more than 10 $\mathrm{km}$) are required. In particular, these large domain heights would be needed for a proper modeling of wind farm induced gravity waves and their impact on blockage effects and production numbers (e.g. Allaerts and Meyers (2017), Lanzilao and Meyers (2022)). Therefore, we performed two sensitivity simulations with increased domain height: one with a height of 6 $\mathrm{km}$ $(2L_z)$ and one with a height of 14.5 $\mathrm{km}$ $(5L_z)$. The results presented in Figs. 20 and 21 do not indicate a significant sensitivity of our results to the domain height (in contrast, explorative model simulations in the early stages of the present study indicated that reducing the domain height to, for instance, 2000 m does have a clear impact on the results).

In addition, Fig. 22 shows the impact of the domain configuration on the ratio of actual to free-stream 140 m wind speeds for wind directions between 15 and 45 degrees. For comparison, the results of the $2L_x$ simulation are cropped to the extent of the REF domain. While the evolution of the wake is comparable to the REF simulation, in the $2L_x$ simulation the flow acceleration along the edges of the wind farm is weaker. The same effect can be seen when the domain height is increased from 3000 to 6000 m $(2L_z)$. Increasing the domain height even further, to 14500 m $(5L_z)$ has a negligible effect on the flow field. This is true, both for the downstream evolution of the wake and the reduction of the wind speed upstream of the wind farm.

The relatively small impact of the domain height reported here may be somewhat surprising given the findings of the studies cited above. However, it could well be that in our study the impact of, for instance, gravity waves is masked by the large variety of synoptic forcings and boundary layer conditions associated with one year of actual weather.

The sensitivity experiments discussed in this Section give a clear indication of the robustness of the presented results: modifying grid spacing, settings of the subgrid model, and extent of the domain within reasonable margins, will likely change the results to several percents at maximum. Overall, we argue that the sensitivity experiments presented here, do not invalidate the reasoning and conclusions discussed in the Results Section.

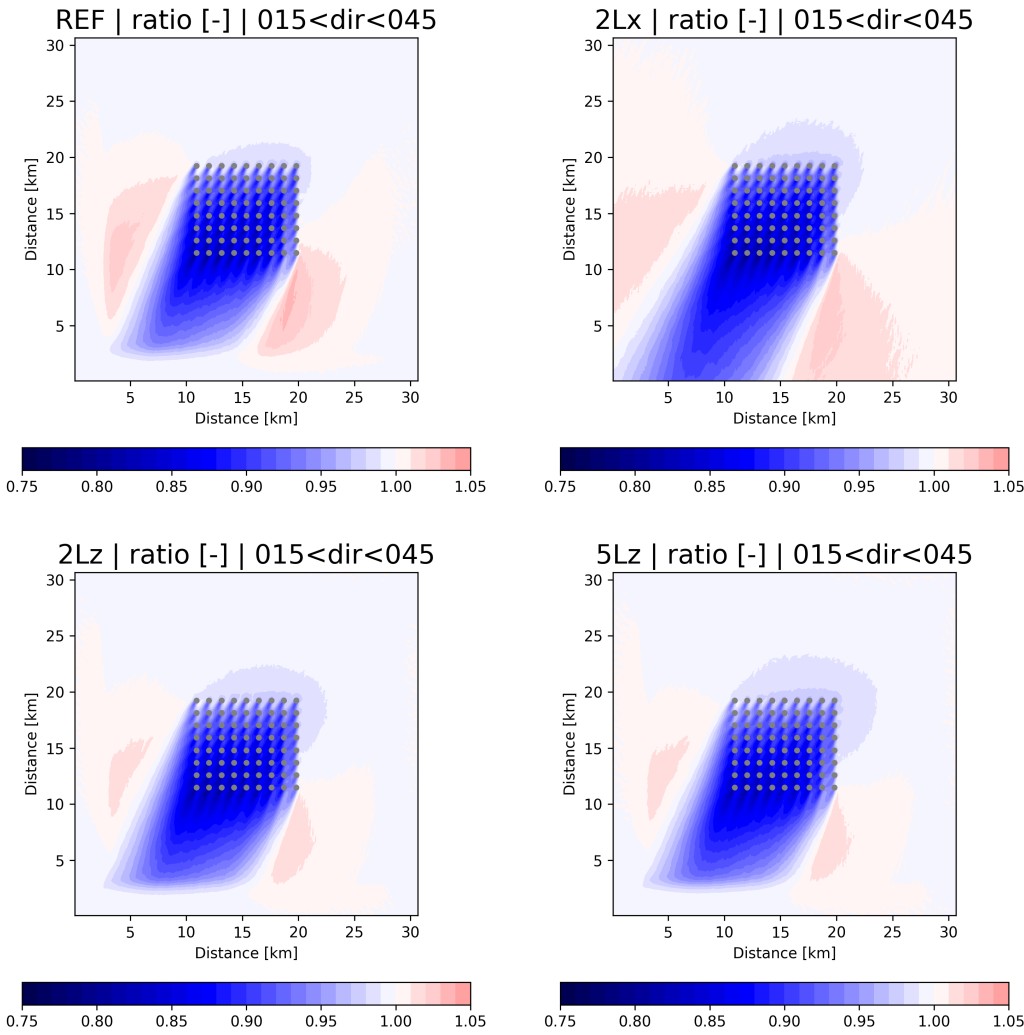

**Figure 22.** Ratio of actual to free-stream 142m wind speed for the REF (a), the $2L_x$ (b), the $2L_z$ (c) and the $5L_z$ (d) IEA10 sensitivity experiments for wind directions between 15 and 45 degrees.

Also, from a broader perspective, the sensitivities described here are not larger than, for instance, sensitivies that are reported in studies with mesoscale models that use wind farm parameterizations like the Fitch et al. (2012) parameterization and/or the

490 explicit wake parameterization of Volker et al. (2015) as discussed in, for example, Pryor et al. (2019) and Fischereit et al. (2022). In addition, engineering models rely on calibration on wind farms with much smaller installed capacities as discussed in the present work and extrapolation to large wind farms is not straightforward. For example, Maas and Raasch (2022) demonstrate that flow dynamics for multi-gigawatt wind farms may differ significantly from smaller-scale wind farms.

## 6 Conclusions

In this work we studied production numbers and aerodynamic losses for six hypothetical 4 GW offshore wind farm scenarios using the GRASP large-eddy simulation model. The six scenarios differed in terms of applied turbine type (eg 2n times 10 MW turbines versus n time 20 MW turbines), installed capacity density (5 MW km$^{-2}$ versus 10 MW km$^{-2}$), and layout. For each scenario, a one-year GRASP simulation was performed using 2015 meteorological large-scale conditions taken from ECMWF's ERA5 reanalysis dataset.

The results suggest that, for the simulated year, aerodynamic losses for a 4 GW offshore wind farm vary from 12 % for 21 MW tubines to 18 % for 10 MW turbines. Moreover, even for turbine types with similar rated capacity but slighty different power and thrust curves, energy production may vary by as much as 7.7 %.

    For all considered scenarios, 80 % of the aerodynamic losses occurs in a narrow wind speed range of 8 to 12 m s$^{-1}$. On the other hand, 50 % of the energy production occurs without any aerodynamic losses, when all turbines operate at rated capacity.

Naturally, these specific numbers should be viewed in context of the considered wind speed probability density function and the wind turbine design choices (power curves).

    Although wind speed is identified as the most important factor determining aerodynamic losses, we were able to isolate the impact of stability. A fair assessment of this impact seemed possible by only considering wind speeds between 6 and 10 m s$^{-1}$. In this wind speed range, aerodynamic losses may be 10 percentage points larger for stably-stratified conditions compared to

510 convective conditions. Numbers vary per scenario with larger differences for scenarios with higher overall losses.

    Losses of first-row turbines, which are related to the global blockage effect, were found to be 2 to 3 % in general. These values are consistent with values of the blockage effect reported in literature. As for the general losses, also the first-row losses occur in a narrow range of disk-averaged wind speed. Also, a clear impact of stability is identified. For example, for disk-averged wind speeds between 6 and 10 m s$^{-1}$, first-row losses may increase to almost 10 % in stably-stratified conditions.

The complexity of disentangling the effect of wind speed and stability is illustrated by considering direction-dependent aerodynamic losses. Only when selecting proper wind speed conditions, a clear impact of stability and of the geometry of the respective scenarios becomes apparent. For instance, when the flow is facing the corners of a square-shaped wind farm, losses are clearly lower than when the flow is directed towards the faces of the wind farm.

    Sensitivity experiments were carried out to better understand the impact of various modelling choices such as resolution and

520 domain height. Results suggest that overall energy production varies with 1 to 2 % depending on model settings and/or the domain configuration. Relative differences between the IEA10MW and Scaled21MW turbine scenario are robust.

In summary, using a high-fidelity modeling technique, the results presented in this explorative study provide a clear indication of the performance of future, multi-gigawatt wind farms for one year of realistic weather conditions. Further research could address several open questions like the influence of the lateral boundary conditions, inter-wind farm wake effects and more validation against meteorological observations and wind farm data. More elaborate validation studies can also shed more light into the resolution dependence of the aerodynamic losses.

*Data availability.* The GRASP large-eddy simulation dataset is available from the authors on request.

*Author contributions.* PB performed the model simulation, analyzed the results and wrote the manuscript. RV, PvD, and HJ provided ideas, corrections and modifications. PvD implemented and calibrated the Actuator Disk Model. HJ is main developer of the GRASP model.

*Competing interests.* The authors have no conflicts of interest to declare that are relevant to the content of this article.

*Acknowledgements.* The authors would like to acknowledge funding from the Topconsortia for Knowledge and Innovation (TKI) funded Winds of the North Sea in 2050 (WINS50) project. The MSc projects of Sebastiaan Ettema and Thijs Bon (Delft University of Technology) on the global-blockage phenomenon provided inspiration and insights for the present work. The constructive comments of the referees are highly appreciated.

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
