# Peer review of "Investigating energy production and wake losses of multi-gigawatt offshore wind farms with atmospheric large-eddy simulation"

_Wind Energy Science, 2022_

## Referee Comment (RC1)

**General comments**

The paper investigates energy production and wake losses of six hypothetical 4 GW offshore wind farm scenarios with a year-long large-eddy simulation. Due to the coupling with a large-scale weather model the dataset represents a large and realistic variety of wind speeds, directions and stabilities that many other LES-studies lack due to their idealized character. The results show in great detail how wake and blockage losses as well as the wind farm wake itself depend on wind speed, wind direction, stability and different wind farm layouts. A sensitivity study provides confidence in the robustness of the results by investigating the effect of e.g. different resolutions and domain sizes. The paper is well structured, the applied methods are comprehensively described and the results are analyzed and discussed thoroughly. In summary, I think that the presented work is very relevant and of great value for the wind energy community.

However, I believe that the LES study has one major shortcoming, that should be addressed: Some recent LES wind farm studies show that the flow through large wind farms is significantly affected by pressure gradients that are related to gravity waves in the stably-stratified free atmosphere which are induced by the wind farm itself (e.g. Allaerts (2017), doi:10.1017/jfm.2017.11, Lanzilao (2022), doi: 10.1088/1742-6596/2265/2/022043 ). The studies show that large domain heights (usually more than 10 km) and a well-tuned damping layer at the domain top is needed to capture these waves properly and to avoid reflection at the upper boundary. Your setup does not meet these requirements and that might distort the results (especially the first-row losses, but also the overall performance and the wind farm wake flow). I guess that a complete rerun of the simulations with a much larger domain height might not be feasible and that the overall effect will be only a few percent. Thus, I only ask you to comment on how this shortcoming might affect the results in the discussion section. A reference to Allaerts et al. (2018) doi:10.1088/1742-6596/1037/7/072006 (Annual impact of wind-farm gravity waves on the Belgian-Dutch offshore wind-farm cluster) might provide some plausibility/verification of your blockage / first-row loss results.

Additionally, I believe that some information needs to be added or corrected in the manuscript. I have commented on the respective parts below.

**Specific comments**

Line 42: Please highlight what the shortcomings of the cited studies are and why there is a need for performing a study like you did.

Line 46: I think that the term "aerodynamic losses" is not a well-known term in the wind energy community and that you have to define it before. Here, in the introduction, you might better explicitly name the two parts of it: wake and blockage losses, which are known terms.

Line 70: How is buoyancy calculated? Is a fixed reference temperature used or rather a vertical profile?

Line 72: "There is a wide variety of subgrid models, of which a significant part uses the concept of eddy viscosity:" Is there a wide variety of SGS models in GRASP or in all LES codes? Please only mention, which model is used in this study.

Line 76: $K_m$ should also be introduced (as eddy viscosity).

Line 81: If the grid is anisotropic, how is the grid spacing defined then? Please add this information.

Line 88: If humidity is part of your prognostic equations, how is condensation and precipitation treated in the model? Most of other LES wind farm studies neglect humidity completely, so please add a few words on this topic.

Line 126: Is the nudging time scale at the boundaries also 6 h or less? I guess that 6 h results in a too weak nudging to damp out the wind farm wake.

Line 135: Which density is taken (anelastic approximation)? The one at hub height?

Line 136: Omit the "²" in $M_D$. Or is it a kind of geometric mean, i.e. the wind speeds in the disc are weighted by squaring?

Line 144: "In order to quantify wake losses, we compare the energy production of the wind turbines with the production of so-called thrustless turbines": You later call these losses "aerodynamic losses" and if I get it right these include wake AND blockage losses. Please use the term "aerodynamic losses" here.

Line 144-150: I wonder whether the thrustless turbines really output the correct power. If they do not exert drag on the flow then also the disc-averaged wind speed and thus power will be higher than for a real (thrustful) turbine (You do not use free-stream wind speed as input). How do you compensate for that?

Line 152: Why the year 2015? Is it a very representative year?

Line 156: "The horizontal domain size extends to 76800 m.": Please make a short reference to the sensitivity study section to show that this is sufficiently large.

Line 165 and Fig. 2: What is meant with the modeled wind speed at 92 m height? Is it a horizontal average over the precursor domain?

Figure 2: Because it is also relevant for your results, a histogram of the stability should be added here. Additionally, I suggest to include a histogram of the boundary layer height, because this is a relevant parameter for wake losses and blockage effect and it is often varied in other LES studies (see my comment on line 373).

Figure 3: Please insert the names of the scenarios as titles of the subplots.

Line 183: I think better than just mentioning the range of rated powers you should list all the used turbines here (name + power).

Line 199: "We designate the difference between the two as 'aerodynamic losses'": Can these losses be further subdivided, e.g. in wake loss and blockage loss? Then please mention here.

Table 1: Please add the names of the scenarios that you use in the figures (e.g. Fig. 5) as a new column.

Line 206: "The aerodynamic losses vary between 12 % and 18 %": Here you provide relative values in % for the aerodynamic losses, but further above (line 199) you define it as a difference (i.e. a power). Please modify your definition accordingly and consider to add an equation for that.

Line 210: "Both the higher production and the higher aerodynamic losses for the IEA10 scenario can be related to behavior of the respective power curves (see Figure 4)." I think the primary reason for the larger power of the IEA 10 turbine is the larger rotor diameter (which results in a higher power levels in region 2 (partial load) of the power curve). Please point out this relation here.

Is it true that the aerodynamic loss is the opposite of what is often called "wind farm efficiency" (loss = 1 - efficiency)? If yes, you should mention it here and also use this term in the abstract and as a keyword because I think that "aerodynamic loss" is not a well-known term in the wind energy community.

Figure 6:

      6 a + d: Are these curves averaged over the entire year?

      6 b + e: This is a density function, so the unit should be GWh / (m/s) or you should mention
          the bin width (in this case a bar plot would be more appropriate)

      6 c + f: Please mention that the values are normalized (although it is obvious). Another option: Do not
          normalize, so that the plot shows the yearly production/loss of each scenario.

Please add this information in the figure caption.

Line 249: "This can be understood by interpreting the total energy production as a function of wind speed as the convolution of the wind speed probability density (Fig. 2b) and the power curves." It is rather a multiplication of the two functions than a convolution, because they are not shifted relative to each other (i.e. the power curve value at a certain wind speed is multiplied with the probability at that same wind speed).

Equation (17): Is the temperature theta_l an absolute temperature in Kelvin or a relative temperature in °Celsius? Based on equation (8) I guess that it is in °C (because 273.15° is subtracted), but it should be an absolute temperature that can

not become negative (refer e.g. to the textbook of Stull "Boundary layer meteorology", chapter and equation 5.6.3, in which the virtual potential temperature is used).

Line 260: See comment on Figure 2 (add histogram of R_b).

Figure 7: - Please add explanation of dashed lines in the caption or in the legend.
How is fractional aerodynamic loss defined and calculated? How is relative loss defined? Is there a difference between relative and fractional loss? If not, please use one term consistently.

Line 267: "Here, for the most stably-stratified conditions, relative losses are roughly 10% larger than for convective conditions." I guess you mean the step from 0.3 to 0.4 between unstable and stable cases. But this is a difference of + 33 % OR 10 percentage **points**. See also line 451 (Conclusions).

Line 297: "Interestingly, the reduction of the first-row wind speed *deficit* (?) continues for much higher wind speeds."

Figure 9: a) please use a nice vertical axis label. b) Please define "first-row loss", here it seems to be normalized (relative loss ?)

Line 344: "The impact of the wind farm on the year-round, omni-directional wind field is in the order of 20 km, after which a velocity deficit of less than 1 % is observed.": Since the domain boundary has only a distance of about 25 km to the wind farm edges, I wonder whether the boundaries affect this result. Or have you also looked at the wakes in the sensitivity study? If not, then please add a critical note here on that.

Line 373: Wind speed, direction and stability are investigated but not boundary layer height. Many other LES wind farm studies vary the boundary layer height and show that it can have a significant effect on the wake and the wind farm performance (e.g. Maas and Raasch, 2022). Thus, I highly recommend to add a subsection and a figure on that topic, if your dataset provides the required information.

Line 409:"In our main simulation, the resolved fraction of the momentum flux is larger than 80 % for 70 % of the time.": At which height have you evaluated the momentum flux?

**Technical corrections**

line 33: "example" instead of "exampe".
line 39: "cluster of offshore wind farms"
line 60: GPUs instead of GPU's
Line 88: Use consistently subgrid or sub-grid (or maybe even better subrid-scale).
Line 110: "enough" instead of "enought"
Line 120: "where alpha is ..." (wrong order)
Line 124: I am not a native speaker, but I would write "setup" instead of "set-up". Please also search for other instances.
Line 125: "Stevens et al." in brackets (use citep instead of citet)
Line 139: "manufacturer" instead of "manufactorer"
Line: 147: thrustless, exert
Line 176 and others: "5 MW.km$^{-2}$" There should be a space between units: " MW km$^{-2}$"
Line 183: "Power and thrust curves"
Line 193: "10 MW/km2" should be "10 MW km$^{-2}$". Please check all other units in the manuscript (line 238, 241,276...)
Line 246: "the impact **of** this effect"
Line 248: "the total energy production peaks at a wind speed of approximately 12 m s$^{-1}$"
Line 257: "over the rotor  of the IEA15 turbine, i.e. between heights of 270 and 30 m"
Line 267: Decide to use "unstable" **OR** "convective" consistently in the entire manuscript (in the figures and the text).
Line 269: "the impact of stab**il**ity"
Line 296: "e.g."
Line 413: "production"
Line 429: "Overall, we argue that the sensitivity experiments presented here, validate the reasoning and conclusions discussed in the Results Section."
Line 473: "The authors would like to acknowledge funding from the Topconsortia for Knowledge and Innovation (TKI) *[missing word here]* funded Winds of the North Sea in 2050 (WINS50) project."

---

## Referee Comment (RC3)

**Comment on *Investigating energy production and wake losses of multi-gigawatt offshore wind farms with atmospheric large-eddy simulation**

January 27 2023

In this paper, by using large-eddy simulation (LES) with coarse grid, the authors examines the energy production and the wake losses of six different hypothetical wind farm, with consideration of atmospheric stability. The novelty of presented in this paper includes setting up boundary conditions by incorporating large scale tendencies from reliable reanalysis dataset and taking transport of moisture and latent heat into account, which make the study presented by the authors stands out among many similar LES studies considering dry ideal conditions only. Moreover, the authors have performed sensitivity analysis which makes their results robust. Thus I recommend publishing this paper.

Several minor or technical comments which I think may help:

- Line 33: typo: exampe -> example

- Line 102-111, section 2.2.1: The authors have talked about how model fields from ERA5 are incorporated to the LES temporally. I was wondering how did the authors spatially incorporate the coarse ERA5 data (which I think have a resolution to the order of 10 km) to the LES run with 120 m resolution. Did the authors interpolate them?

- Line 134 - 137: It seems that the authors have incorporated a simple ADM without wind farm rotation. Maybe out of scope, but I was wondering if the results will be different if rotation has been accounted for. There are several ADMs that already considered rotation maybe the authors can check those.

- Line 139: typo: manufactorer -> manufacturer

- Line 155-160: The resolution of the LES run:

  - Could the authors provide more information about the vertical grid stretching thus the vertical resolution across the rotor?

– The 120 m horizontal grid spacing (and the 60 m in the sensitivity test part later) is very interesting since it sits between the resolution of mesoscale models with wind turbine modeled (e.g. WRF with Fitch model where the resolution is in the order of 1 kilometer) and that of the wake-resolving LES (in the order of 1 meter). Since there is no resolving of the wake in this study, the coarse resolution makes sense. I was wondering apart from the observations and wind farm data, have the authors also considered comparing the results with those from mesoscale models and wake-resolving LES? This may be a worthy point in future studies.

- Line 258: Why do the authors chose 270 m and 30 m to calculate the bulk Richardson number?

- Overall comment

  – I think the authors should format the reference to the Figures in the text like Figure X a), b), c), etc. to be consistent with the sub figures in the paper. Currently they are in a format like Figure X a, b, c, ...

  – Just curious, have the authors considered seasonal variations for this year-round LES study? No LES studies so far have done that but there are other large-scale studies that focuses on seasonal changes of wind farm wake losses and production (e.g. Pryor et al. 2018, Wan et al. 2012). Maybe it will be interesting to consider seasonal effects.

---

## Author Comment (AC1)

We thank the reviewer for his/her time and efforts in carefully reading our manuscript. The feedback on our work is highly appreciated!

A point-by-point reply to the individual comments can be found below in blue (the original comments are included in black).

This paper investigates the energy production and wake losses of multi-gigawatt offshore wind farms using the LES approach. I like the research idea and the approach of using GPU to perform LES simulations with real atmospheric-driven forcings. I would support publication after the authors addressing these minor comments.

- Line 212: Please make a table showing the total number of wind turbines for each wind farm scenarios (Figure 3).

The number of wind turbine for each scenario is included in Table 1

- Figure 5: Can the authors better explain how they calculate the free-stream production and the actual production? I would imagine the free-steam production (Figure 5a) of the first 4 scenarios to be 42TWh. However, the figure only shows about half of that magnitude.

Free-stream of gross production is the production that would occur in the absence of wake and blockage effect but with taking into account the actual wind speed distribution. As the ambient wind speed is frequently lower than the rated wind speed of the turbines, the free-stream production is considerably lower than the production based on the rated capacity of 4GW (for the first 4 scenarios a rated production would amount to 4.2GW x (24 x 365)h = 37TWh). In contrast, the free-stream production is lower, namely around 22TWh

- I would assume that the layout of the wind farm would have a substantial impact on the power production and the aerodynamic loss (Figure3). Do the authors find any significant differences in the result between scenario 6 and the rest.

The differences in terms of overall (year-round) aerodynamic losses are small. When considering dependence on wind direction a clear impact is seen, see Section 4.4 'Directional effects'

- Why are the turbine spacings in these scenarios different (Table1). Shouldn't the authors just change the wind turbine type while keeping everything else the same?

This depends on the research question that is to be answered. In our case we keep the total capacity (4GW) and the installed capacity density (10 MW/km2) the same. This implies that when the turbine type and/or the number of turbines is changed also the spacing in term of rotor diameter may become different.

---

## Author Comment (AC2)

We thank the reviewer for his/her time and efforts in carefully reading our manuscript. The feedback on our work is highly appreciated!

A point-by-point reply to the individual comments can be found below in blue (the original comments are included in black).

Comment on Investigating energy production and wake losses of multi-gigawatt offshore wind farms with atmospheric large-eddy simulation

January 27 2023

In this paper, by using large-eddy simulation (LES) with coarse grid, the authors examines the energy production and the wake losses of six different hypothetical wind farm, with consideration of atmospheric stability. The novelty of presented in this paper includes setting up boundary conditions by incorporating large scale tendencies from reliable reanalysis dataset and taking transport of moisture and latent heat into account, which make the study presented by the authors stands out among many similar LES studies considering dry ideal conditions only. Moreover, the authors have performed sensitivity analysis which makes their results robust. Thus I recommend publishing this paper.

Several minor or technical comments which I think may help:
• Line 33: typo: exampe -> example

Corrected

• Line 102-111, section 2.2.1: The authors have talked about how model fields from ERA5 are incorporated to the LES temporally. I was wondering how did the authors spatially incorporate the coarse ERA5 data (which I think have a resolution to the order of 10 km) to the LES run with 120 m resolution. Did the authors interpolate them?

Clarified by adding: 'Initial conditions and large-scale (*LS*) tendencies are extracted from ERA5 by means of spatial and temporal interpolation and prescribed to GRASP as a function of height only (i.e. homogeneous over the domain).'

• Line 134 - 137: It seems that the authors have incorporated a simple ADM without wind farm rotation. Maybe out of scope, but I was wondering if the results will be different if rotation has been accounted for. There are several ADMs that already considered rotation maybe the authors can check those.

We have no experience with ADMs that consider rotation. From that, it's hard to make any statement how using these models will impact the results. As the number of turbines is large in the simulated 4GW wind farms, the impact may be small, but this is mere speculation. We do have experience with (rotating) actuator line models, but this would require a totally different study in terms of resolution (in fact, simulation of the present domain with the then required resolution is computationally unfeasible). And, as such, a comparison with the present setup is hard to make.

No changes made.

• Line 139: typo: manufactorer -> manufacturer

Corrected

• Line 155-160: The resolution of the LES run:

– Could the authors provide more information about the vertical grid stretching thus the vertical resolution across the rotor?

Added: The number of vertical levels is already indicated: 48. Grid stretching was done by applying a uniform growth factor from the lowest model level upwards. Added: '(i.e. a uniform growth factor of 2.845 %)' (See also comment on line 156 RC2)

– The 120 m horizontal grid spacing (and the 60 m in the sensitivity test part later) is very interesting since it sits between the resolution of mesoscale models with wind turbine modeled (e.g. WRF with Fitch model where the resolution is in the order of 1 kilometer) and that of the wake-resolving LES (in the order of 1 meter). Since there is no resolving of the wake in this study, the coarse resolution makes sense. I was wondering apart from the observations and wind farm data, have the authors also considered comparing the results with those from mesoscale models and wake-resolving LES? This may be a worthy point in future studies.

Indeed, no comparison with other models has been made for the present study. For sure, it would be interesting to compare the present results with those from mesoscale models (requires careful alignment of scenarios and simulations though, to make meaningful comparison).

• Line 258: Why do the authors chose 270 m and 30 m to calculate the bulk Richardson number?

This these two heights correspond to the highest and lowest point of the rotor of the IEA15 turbine ($z\_hub=150m$, $r=120m$). This is indicated in the text.

• Overall comment
– I think the authors should format the reference to the Figures in the text like Figure X a), b), c), etc. to be consistent with the sub figures in the paper. Currently they are in a format like Figure X a, b, c, ...
For now, we choose to stick with the original way of referring to the Figures

– Just curious, have the authors considered seasonal variations for this year-round LES study? No LES studies so far have done that but there are other large-scale studies that focuses on seasonal changes of wind farm wake losses and production (e.g. Pryor et al. 2018, Wan et al. 2012). Maybe it will be interesting to consider seasonal effects.

We have not checked this, but for practical purposes this is relevant indeed (although no clear conclusions on seasonal effects can be drawn from a single year). However, seasonal variations will be induced by variations in more physical properties like wind speed distribution and stability. So indirectly our results give an indication what will happen in windy seasons with unstable conditions versus maybe seasons with a lot of stably stratified conditions and calmer winds.

---

## Author Comment (AC3)

We thank the reviewer for his/her time and efforts in carefully reading our manuscript. The feedback on our work is highly appreciated!

A point-by-point reply to the individual comments can be found below in blue (the original comments are included in black).

This work used a year-long LES simulation dataset to looks at energy production and wake losses for six different 4 GW offshore wind farm scenarios. Authors looked at wake and losses for six different scenarios as a function of wind speed and direction, stability, as well as different wind farm layouts.

I believe that paper is well structured, and this work is very relevant for the entire wind energy community.

 Some specific comments:

Line 72 and 77: decide if you would like to use "subgrid models" or "subgrid-models" and be consistent.

Choose 'subgrid model'.

Line 80: Km is not defined

Corrected

Line 83: "eddy-viscosity model specifically developed for anisotropic grids. " So the grid is anisotropic? Need more details about grid spacing.

No statements about the grid configuration are made here (the grid configuration /simulation set-up itself is discussed in Section 2.4). Here, the reference to anisotropic grid just refers to one of the characteristics of the applied Rozema subgrid scheme. (see also reply to comment Line 81 RC1)

Line 88: So this model includes moisture and phase changes? How ere clouds and precipitations treated in the model? Need more information about moist processes in the model.

Requested information has been added (see also reply to comment on Line 88 by  RC1).

Line 152: Why you picked 2015? Is there any particulate reason? Need additional info about this choice.

2015 was chosen because of availability of Meteomast IJmuiden observations. Added this motivation directly at this location in the text instead of a couple of lines later. (similar to comment Line 152 RC1)

Line 156: What type of "Vertical grid stretching"? How many points there is in Z direction and how is greed spacing distributed? Need additional clarification.

The number of vertical levels is already indicated: 48. Grid stretching was done by applying a uniform growth factor from the lowest model level upwards. Added: '(i.e. a uniform growth factor of 2.845 %)'

Very important note here (regarding model setup): Many studies reported that wind farm generates gravity waves (for example Allaerts (2017)), and I'm not sure that authors accounted for that fact. Without proper treatment of these waves, they might affect overall result, so should be properly addressed. Authors should address/discuss absence of proper treatment of farm induced gravity waves and their possible impact on this whole analysis.

By nature, the LES that we use is capable of representing wind farm induces gravity waves. One can debate to what extent our simulation setup allows for proper modeling of these waves. We added results of an additional sensitivity simulation with a 14.5km high domain and added a discussion on this topic.

See reply to main general comment RC1 on this topic.

Line 156: Is there a reason for a domain to be horizontally 76800m long?

It is not smaller because then it would be too small for modeling the designed 4GW wind farms, it is not larger because of computational costs. Added: 'Sensitivity experiments discussed in Section 5 indicate that this domain size is sufficiently large.' (see also comment on line 156 RC1)

Line 157: what about upper boundary comditions?

Good point. Added a subsection 'Upper boundary conditions'

Line 293: "Schneemann et al. (2021), . " delete ,

Corrected

Line 341: "averaged over all the entire year" delete "all"

Done

---

## Author Comment (AC4)

We thank the reviewer for his time and effort in carefully reading our manuscript. The detailed feedback of the reviewer on our work is much appreciated!

A point-by-point reply to the individual comments can be found below in blue (the original comments are included in black).

Following the reviewer's comments two major modifications have been made:

- The impact of the domain height is discussed, including an additional sensitivity study with a 14.5 km high domain.
- An analysis of the impact of the boundary layer height has been added.

General comments
The paper investigates energy production and wake losses of six hypothetical 4 GW offshore wind farm scenarios with a year-long large-eddy simulation. Due to the coupling with a large-scale weather model the dataset represents a large and realistic variety of wind speeds, directions and stabilities that many other LES-studies lack due to their idealized character. The results show in great detail how wake and blockage losses as well as the wind farm wake itself depend on wind speed, wind direction, stability and different wind farm layouts. A sensitivity study provides confidence in the robustness of the results by investigating the effect of e.g. different resolutions and domain sizes. The paper is well structured, the applied methods are comprehensively described and the results are analyzed and discussed thoroughly. In summary, I think that the presented work is very relevant and of great value for the wind energy community.

However, I believe that the LES study has one major shortcoming, that should be addressed: Some recent LES wind farm studies show that the flow through large wind farms is significantly affected by pressure gradients that are related to gravity waves in the stably-stratified free atmosphere which are induced by the wind farm itself (e.g. Allaerts (2017), doi:10.1017/jfm.2017.11, Lanzilao (2022), doi: 10.1088/1742-6596/2265/2/022043 ). The studies show that large domain heights (usually more than 10 km) and a well-tuned damping layer at the domain top is needed to capture these waves properly and to avoid reflection at the upper boundary. Your setup does not meet these requirements and that might distort the results (especially the first-row losses, but also the overall performance and the wind farm wake flow). I guess that a complete rerun of the simulations with a much larger domain height might not be feasible and that the overall effect will be only a few percent. Thus, I only ask you to comment on how this shortcoming might affect the results in the discussion section. A reference to Allaerts et al. (2018) doi:10.1088/1742-6596/1037/7/072006 (Annual impact of wind-farm gravity waves on the Belgian-Dutch offshore wind-farm cluster) might provide some plausibility/verification of your blockage / first-row loss results.

To start with, a subsection has been added that describes the upper boundary conditions (Section 2.2.3). This information was lacking in the original manuscript and is relevant for the present discussion. It mentions, for example, the fact that we apply a sponge layer in the upper quarter of the domain to damp out waves and prevent those from reflecting at the upper boundary.

Then, to accommodate reviewer's concern, an additional 100-day sensitivity simulation with a domain height of 14.5 km has been performed. The results are added to the Discussion Section.

In addition to the Figures that were already present, an extra Figure has been added to give a spatial impression of the impact of domain height (Lz=3000, 6000, 14500m) and horizontal extent (Lx and 2Lx) on the evolution of wake / blockage patterns.

The results of the sensitivity simulations indicate only modest sensitivity of our results to the domain height.

Partly, this may be because even without considering gravity waves a (large) wind farm will induce significant horizontal pressure gradients over the wind farm by continuity arguments (which may even increase in strength in case of too low domains).

Another reason may be that in our study the impact of gravity waves is masked by the complex mix of synoptic and boundary layer conditions associated with one year of realistic weather.

This is not to say that our domain configuration cannot be improved (ideally, the domain would be larger, higher and with finer resolution). The results presented in the Discussion Section suggest the present setup provides robust results and strikes at least a reasonable balance between accuracy and computational costs.

References to studies cited above have been made. Because of differences in approach, definitions, and applied models, a direct comparison of numbers (eg blockage losses) is hard to make. Therefore we suffice with noting that "although the applied definitions and metrics can be discussed, these (i.e. the here presented) values are not inconsistent with values of the blockage effect reported in literature). "

Additionally, I believe that some information needs to be added or corrected in the manuscript. I have commented on the respective parts below.

Specific comments
Line 42: Please highlight what the shortcomings of the cited studies are and why there is a need for performing a study like you did.

The motivation for performing the present study in relation to earlier work is described in the next paragraph. Following the reviewer's comment, the formulation of this phrase was sharpened a bit: "As such, the present work allows for a more statistical approach to study wind farm dynamics compared to other LES studies that mostly considered a set of idealized case studies."

Line 46: I think that the term "aerodynamic losses" is not a well-known term in the wind energy community and that you have to define it before. Here, in the introduction, you might better explicitly name the two parts of it: wake and blockage losses, which are known terms.

Changed "aerodynamic losses" into "wake and blockage losses"

Line 70: How is buoyancy calculated? Is a fixed reference temperature used or rather a vertical profile?

The latter. Added: "In the buoyancy calculation a height-dependent reference temperature is used."

Line 72: "There is a wide variety of subgrid models, of which a significant part uses the concept of eddy viscosity:" Is there a wide variety of SGS models in GRASP or in all LES codes? Please only mention, which model is used in this study.

Done. We now just introduce the model that is used in this study (the Rozema model).

Line 76: $K_m$ should also be introduced (as eddy viscosity).

Done

Line 81: If the grid is anisotropic, how is the grid spacing defined then? Please add this information.

No statements about the grid configuration are made here (the grid configuration /simulation set-up itself is discussed in Section 2.4). Here, the reference to anisotropic grid just refers to one of the characteristics of the applied Rozema subgrid scheme. (similar to comment Line 83 RC2)

Line 88: If humidity is part of your prognostic equations, how is condensation and precipitation treated in the model? Most of other LES wind farm studies neglect humidity completely, so please add a few words on this topic.

Some more details about the cloud formulation and microphysics are added:

"An 'all-or-nothing' cloud adjustment scheme is used that assumes that no cloud water/ice is present in unsaturated grid boxes, while all moisture exceeding the local saturated vapor pressure is considered liquid water or ice. In addition, the Grabowski (1998) ice microphysics scheme is used. A single precipitating prognostic variable $q_r$ is used. The partitioning towards water, snow and graupel is diagnosed with a temperature criterion. Autoconversion, the initial stage of rain drop formation, is modeled according the Kessler-Lin formulation (Khairoutdinov and Randall 2003)."

Line 126: Is the nudging time scale at the boundaries also 6 h or less? I guess that 6 h results in a too weak nudging to damp out the wind farm wake.

Indeed, the nudging at the boundaries of the actual simulation towards the precursor simulation is much stronger than the overall 6-hour nudging towards the ERA5 boundary conditions. It serves a completely different purpose. This is clarified in the text: 'Over a boundary region the values of the 'actual' simulation are strongly nudged towards the precursor simulation (with an adaptive nudging time scale in the order of the model time step)'.

Line 135: Which density is taken (anelastic approximation)? The one at hub height?

For the power and thrust calculations the disk-averaged density is taken. Clarified

Line 136: Omit the "²" in M_D. Or is it a kind of geometric mean, i.e. the wind speeds in the disc are weighted by squaring?

Omitted the 2, typo. Thanks for catching!

Line 144: "In order to quantify wake losses, we compare the energy production of the wind turbines with the production of so-called thrustless turbines": You later call these losses "aerodynamic losses" and if I get it right these include wake AND blockage losses. Please use the term "aerodynamic losses" here.

Changed

Line 144-150: I wonder whether the thrustless turbines really output the correct power. If they do not exert drag on the flow then also the disc-averaged wind speed and thus power will be higher than for a real (thrustful) turbine (You do not use free-stream wind speed as input). How do you compensate for that?

Well-noticed, this is not trivial indeed. As explained in the original manuscript, for the 'actual' turbines disc-based power and thrust coefficients (power curves) were obtained by means of an offline simulation. The same procedure is followed for the thrustless turbines, but now with the Ct curve set to 0 in the corresponding offline simulation. With the resulting 'thrustless disk-based powercurves' the thrustless turbines do produce the correct power.

Added: "The disk-based power coefficients for the thrustless turbines are obtained by means of a separate offline simulation with the thrust coefficients set to 0."

Line 152: Why the year 2015? Is it a very representative year?

2015 was chosen because of availability of Meteomast IJmuiden observations. Added this motivation directly at this location in the text instead of a couple of lines later.

Line 156: "The horizontal domain size extends to 76800 m.": Please make a short reference to the sensitivity study section to show that this is sufficiently large.

Added: 'Sensitivity experiments discussed in Section 5 indicate that this domain size is sufficiently large.'

Line 165 and Fig. 2: What is meant with the modeled wind speed at 92 m height? Is it a horizontal average over the precursor domain?

Clarified by adding 'In this case, the modeled (horizontal) wind speed is taken from a virtual metmast placed at the location of the actual metmast.'

Figure 2: Because it is also relevant for your results, a histogram of the stability should be added here. Additionally, I suggest to include a histogram of the boundary layer height, because this is a relevant parameter for wake losses and blockage effect and it is often varied in other LES studies (see my comment on line 373).

Adding a histogram of stability was considered for the original manuscript, but it was deliberately left out for the following reasons:

- In our case, a histogram of stability would be a histogram of the bulk Richardson number between 270 and 30m. Although this quantity allows for a proper classification in broad stability classes, the absolute value of Rb is ambiguous. For instance, taking another height interval would lead to considerable different values (most obvious when taking the surface as the lower level). As such, translation to other studies and/or applications is not trivial.
- Other measures of stability like the surface Obukhov length, may even give a significant change in the number of stable versus unstable cases (surface quantity versus bulk quantity over a relatively deep layer).
- Relevance of a histogram for the presented results is low as we only use Rb to define broad classes of stability; no further reference to the histograms would be made in the main text.
- The presented values of the 33th and 66th percentiles already give some indication of the Rb distribution.
- An analysis of the impact of boundary layer height is added (see reply2comment line373). But also here, we choose to work with broad classes (low, medium and high boundary layers) and leave out a histogram and provide information on the distribution in a similar way as for the bulk Richardson number.

Given the above, we are not convinced of the added value of the histograms.

Figure 3: Please insert the names of the scenarios as titles of the subplots.

Done.

Line 183: I think better than just mentioning the range of rated powers you should list all the used turbines here (name + power).

Rephrased to 'To study the impact of using different turbine types while keeping the total installed power approximately the same, four different turbine types have been applied. Three reference wind

turbines were used with data taken from https://nrel.github.io/turbine-models/Offshore.html: the DTU_10MW_178RWT turbine (10.6 MW, labeled as DTU10), the IEA_10MW_198RWT turbine (10.6 MW, labeled as IEA10), and the IEA_15MW_240RWT turbine (15 MW, labeled as IEA15).'

Line 199: "We designate the difference between the two as 'aerodynamic losses'": Can these losses be further subdivided, e.g. in wake loss and blockage loss? Then please mention here.

Separating wake and blockage effects is virtually impossible. Therefore, we prefer the more generic term 'aerodynamic losses.'.

Table 1: Please add the names of the scenarios that you use in the figures (e.g. Fig. 5) as a new column.

Added a column 'label' with the scenario names used in the Figures. Removed the column 'short' with the turbine abbreviations (defined in the main text now and overlap with scenario labels).

Line 206: "The aerodynamic losses vary between 12 % and 18 %": Here you provide relative values in % for the aerodynamic losses, but further above (line 199) you define it as a difference (i.e. a power). Please modify your definition accordingly and consider to add an equation for that.

We clarified the definition of the term aerodynamic losses:

'We distinguish between production of the thrustless turbines (also called `free-stream production' or `gross power') and the actual production (`net power'). We designate the difference between the two as the `aerodynamic losses'. Depending on the application, we present either absolute aerodynamic losses (in MW or MWh) or relative aerodynamic losses (dimensionless) where the absolute losses are normalized with the free-stream production.'

Line 210: "Both the higher production and the higher aerodynamic losses for the IEA10 scenario can be related to behavior of the respective power curves (see Figure 4)." I think the primary reason for the larger power of the IEA 10 turbine is the larger rotor diameter (which results in a higher power levels in region 2 (partial load) of the power curve).
Please point out this relation here.

Indeed the rotor diameter plays a role. But only in combination with the respective power curves. Added 'rotor diameter' as a cause for the different production/losses: '... can be related to a difference in the rotor diameter and a different behavior of the respective power curves.'

Is it true that the aerodynamic loss is the opposite of what is often called "wind farm efficiency" (loss = 1 - efficiency)? If yes, you should mention it here and also use this term in the abstract and as a keyword because I think that "aerodynamic loss" is not a well-known term in the wind energy community.

The use of the term 'wind farm efficiency' is ambiguous: it is used for the production with respect to the rated/installed power but sometimes also for the production with respect to the free stream power. Therefore, we choose to stay away from this term.

Figure 6:
6 a + d: Are these curves averaged over the entire year?

Yes, clarified in the text, added to caption.

6 b + e: This is a density function, so the unit should be GWh / (m/s) or you should mention the bin width (in this case a bar plot would be more appropriate)

Added the bin-width of 1m/s in the caption. Indeed, formally these are histograms, but for practical reasons we present those as line plots.

6 c + f: Please mention that the values are normalized (although it is obvious). Another option: Do not normalize, so that the plot shows the yearly production/loss of each scenario.
Please add this information in the figure caption.

Done. We stick to normalized data here in order to the make the point that 50 of production occurs above 14m/s and 80% of the losses between 8 and 12m/s.

Line 249: "This can be understood by interpreting the total energy production as a function of wind speed as the convolution of the wind speed probability density (Fig. 2b) and the power curves." It is rather a multiplication of the two functions than a convolution, because they are not shifted relative to each other (i.e. the power curve value at a certain wind speed is multiplied with the probability at that same wind speed).

Changed convolution -> multiplication

Equation (17): Is the temperature theta_l an absolute temperature in Kelvin or a relative temperature in °Celsius? Based on equation (8) I guess that it is in °C (because 273.15° is subtracted), but it should be an absolute temperature that can not become negative (refer e.g. to the textbook of Stull "Boundary layer meteorology", chapter and equation 5.6.3, in which the virtual potential temperature is used).

This formulation was indeed inconsistent. We choose to remove the temperature offset from the model description.

Line 260: See comment on Figure 2 (add histogram of R_b).

See reply on Figure 2 above

Figure 7: - Please add explanation of dashed lines in the caption or in the legend.

Done

How is fractional aerodynamic loss defined and calculated? How is relative loss defined? Is there a difference between relative and fractional loss? If not, please use one term consistently.

Good point. Replaced 'fractional loss' by 'relative loss' throughout the manuscript.

Line 267: "Here, for the most stably-stratified conditions, relative losses are roughly 10% larger than for convective conditions." I guess you mean the step from 0.3 to 0.4 between unstable and stable cases. But this is a difference of + 33 % OR 10 percentage points. See also line 451 (Conclusions).

Changed % to 'percentage points' both at this place and in the conclusions.

Line 297: "Interestingly, the reduction of the first-row wind speed deficit (?) continues for much higher wind speeds."

Changed into: 'Interestingly, the first-row wind speed deficit with respect to free-stream conditions continues towards much higher wind speeds.'

Figure 9: a) please use a nice vertical axis label.

Changed y-axis label to 'Wind speed deficit [m/s]'

b) Please define "first-row loss", here it seems to be normalized (relative loss ?)

Correct, it's a relative loss here. Added to Caption.

In addition, panel 9c was removed as exactly the same panel is part of Fig 7.

Line 344: "The impact of the wind farm on the year-round, omni-directional wind field is in the order of 20 km, after which a velocity deficit of less than 1 % is observed.": Since the domain boundary has only a distance of about 25 km to the wind farm edges, I wonder whether the boundaries affect this result. Or have you also looked at the wakes in the sensitivity study? If not, then please add a critical note here on that.

We added a figure to the Discussion section that shows the wake evolution for a selected 30-degree wind direction sector. It shows that the dependence on domain height and horizontal extent is small. Or better formulated, it indicates that the applied domain configuration had no obvious adverse impact on the results.

Line 373: Wind speed, direction and stability are investigated but not boundary layer height. Many other LES wind farm studies vary the boundary layer height and show that it can have a significant effect on the wake and the wind farm performance (e.g. Maas and Raasch, 2022). Thus, I highly recommend to add a subsection and a figure on that topic, if your dataset provides the required information.

An analysis to the impact of boundary layer height has been added to Section 4.2 which is renamed to 'Impact of stability and boundary layer height'. In short:

- We diagnosed the boundary layer height from the precursor model output. We define the boundary layer height as the height at which the magnitude of the vertical momentum flux becomes less than 5% of its surface value.
- We determined the aerodynamic losses as a function of wind speed for tertiles of boundary layer height (in a similar way as was done for stability). Discussion plus Figure added.
- Within the windspeed range of 6-10m/s. We found for the lowest 33% boundary layers (h<341m) 10 %-point higher aerodynamic losses than for the deepest 33% boundary layers (h>955m).
- For stronger wind speeds impact boundary layer is small (rated power, no losses), for weaker wind speeds as well (low power anyway).
- The simultaneous occurrence of the classes of stability and boundary layer height is shown in a contingency table.

Line 409:"In our main simulation, the resolved fraction of the momentum flux is larger than 80 % for 70 % of the time.": At which height have you evaluated the momentum flux?

Added the answer to the text: 'at a height of 150 m, which is the hub-height of the IEA15 turbine'

Technical corrections
line 33: "example" instead of "exampe". Changed
line 39: "cluster of offshore wind farms" Changed
line 60: GPUs instead of GPU's Changed
Line 88: Use consistently subgrid or sub-grid (or maybe even better subrid-scale).

Changed to subgrid-scale

Line 110: "enough" instead of "enought" Changed
Line 120: "where alpha is ..." (wrong order) Changed
Line 124: I am not a native speaker, but I would write "setup" instead of "set-up". Please also search for other instances. Tricky one, changed to setup after studying https://grammarist.com/spelling/set-up-vs-setup/ ;)

Line 125: "Stevens et al." in brackets (use citep instead of citet) Changed
Line 139: "manufacturer" instead of "manufactorer" Changed
Line: 147: thrustless, exert Changed
Line 176 and others: "5 MW.km ²" There should be a space between units: " MW km ²"⁻ ⁻ Changed
Line 183: "Power and thrust curves" Changed
Line 193: "10 MW/km2" should be "10 MW km ²". Please check all other units in the manuscript (line 238, 241,276...)⁻ Changed, checked
Line 246: "the impact of this effect" Changed
Line 248: "the total energy production peaks at a wind speed of approximately 12 m s ¹"⁻ Changed
Line 257: "over the rotor blade of the IEA15 turbine, i.e. between heights of 270 and 30 m" Unclear, no change made

Line 267: Decide to use "unstable" OR "convective" consistently in the entire manuscript (in the figures and the text). Changed all 'unstable' into 'convective'
Line 269: "the impact of stability" Changed
Line 296: "e.g." Changed
Line 413: "production" Changed
Line 429: "Overall, we argue that the sensitivity experiments presented here, do no invalidate the reasoning and conclusions discussed in the Results Section." Changed no -> not
Line 473: "The authors would like to acknowledge funding from the Topconsortia for Knowledge and Innovation (TKI) [missing word here] funded Winds of the North Sea in 2050 (WINS50) project." No missing words found, no changes made.